

# The size resolved cloud condensation nuclei (CCN) activity and its prediction based on aerosol hygroscopicity and composition in the Pearl Delta River (PRD) Region during wintertime 2014

Mingfu Cai[1,2], Haobo Tan[2*], Chak K. Chan[3], Yiming Qin[4,5], Hanbing Xu[1], Fei Li[2],

Misha I. Schurman[4], Liu Li[1], and Jun Zhao[1*]

[1] School of Atmospheric Sciences, Guangdong Province Key Laboratory for Climate Change and Natural Disaster Studies, and Institute of Earth Climate and Environment System, Sun Yat-sen University, Guangzhou, Guangdong 510275, China

[2] Institute of Tropical and Marine Meteorology/Guangdong Provincial Key Laboratory of Regional Numerical Weather Prediction, CMA, Guangzhou 510640, China

[3] School of Energy and Environment, City University of Hong Kong, Hong Kong, China

[4] Hong Kong University of Science and Technology, Hong Kong, China

[5] School of Engineering and Applied Sciences, Harvard University, Cambridge, Massachusetts 02138, United States

*Corresponding authors: Haobo Tan (hbtan@grmc.gov.cn) and Jun Zhao (zhaojun23@mail.sysu.edu.cn)

**Abstract.** A hygroscopicity-tandem differential mobility analyzer (H-TDMA), a scanning mobility CCN analyzer (SMCA), and an aerodyne high resolution time-of-flight aerosol mass spectrometer (HR-ToF-AMS) were used to respectively measure the hygroscopicity, condensation nuclei activation, and chemical composition of aerosol particles at the Panyu site in the Pearl River Region during wintertime 2014. The distribution of the size-resolved cloud condensation nuclei (CCN) at four supersaturations (SS=0.1%, 0.2%, 0.4%, and 0.7%) and the aerosol particle size distribution were obtained by the SMCA. The hygroscopicity parameter κ ($\kappa_{CCN}$, $\kappa_{H-TDMA}$, and $\kappa_{AMS}$) was respectively





calculated based upon the SMCA, H-TMDA, and AMS measurements. The results showed that the
$\kappa_{H\text{-}TDMA}$ value was slightly smaller than the $\kappa_{CCN}$ one at all diameters and for particles larger than 100
nm the $\kappa_{AMS}$ value was significantly smaller than the others ($\kappa_{CCN}$, and $\kappa_{H\text{-}TDMA}$), which could be
attributed to the underestimated hygroscopicity of the organics ($\kappa_{org}$). The activation ratio (AR)
calculated from the growth factor – probability density function (Gf-PDF) without surface tension
correction was found to be lower than that from the H-TDMA measurement, due most likely to the
uncorrected surface tension ($\sigma_{s/a}$) that did not consider the surfactant effects of the organic compounds.
We demonstrated that better agreement between the calculated and measured AR could be obtained by
adjusting $\sigma_{s/a}$. Various schemes were proposed to predict the CCN number concentration ($N_{CCN}$) based
on H-TDMA and AMS measurements. In general, the predicted $N_{CCN}$ agreed reasonably well with the
corresponding measured ones using different schemes. For H-TDMA measurements, the $N_{CCN}$ value
predicted from the real time AR measurements was slightly smaller (~6.8%) than that from the
activation diameter ($D_{50}$) method due to the assumed internal mixing in the $D_{50}$ prediction. The $N_{CCN}$
values predicted from bulk $PM_1$ were higher (~11.5%) than those from size-resolved composition
measured by the AMS because a significant fraction of $PM_1$ was composed of inorganic matter. The
$N_{CCN}$ calculated from AMS measurement were under-predicted at 0.1% and 0.2% supersaturations,
which could be due to underestimate of $\kappa_{org}$ and overestimate of $\sigma_{s/a}$. For SS=0.4% and 0.7%, slight
over-predicted $N_{CCN}$ was found because of the internal mixing assumption. Our results highlight the
need for accurately evaluating the effects of organics on both the hygroscopic parameter $\kappa$ and the
surface tension $\sigma$ in order to accurately predict CCN activity.



**1 Introduction**
Aerosol particles can directly impact global climate by scattering and absorbing solar radiation
(Stocker, 2013), while they can influence cloud formation, life time and optical properties by acting as
cloud condensation nuclei (CCN), indirectly exerting climatic forcing on the Earth's atmosphere. In
general, aerosol particles increase the CCN concentration and hence cause cooling effects on the global
radiation balance. However, to what extent aerosol particles contribute to the radiation forcing is still
highly uncertain (Stocker, 2013). It is hence important to measure chemical composition and properties
of aerosol particles in order to assess their abilities of acting as CCN and contribution to cloud
formation, further facilitating our understanding of the impacts of atmospheric aerosols on regional and
global climate.
The extent to which aerosol particles can affect cloud formation is dependent on their fraction that can
be activated to become CCN. This fraction of activation is termed as CCN activity that is determined
by the chemical composition, sizes, and the water saturation ratio of the particles (Farmer et al., 2015).
The size-dependent saturation ratio (S) can be calculated from the Köhler equation (Köhler, 1936):
$$S = a_w \exp\left(\frac{4\sigma_{s/a} M_w}{RT \rho_w D}\right) \tag{1}$$
where $a_w$ is the water activity in solution, $\sigma_{s/a}$ is surface tension of the solution/air interface, $M_w$ is
the mole weight of water, R is the universal gas constant, T is temperature in Kelvin, and D is the
diameter of the droplet. The $a_w$ represents Raoult effect, which means that the activation potential
increases with the concentration of the solution. The term $\exp\left(\frac{4\sigma_{s/a} M_w}{RT \rho_w D}\right)$ represents Kelvin effect,
which relates the surface curvature to the saturation vapor pressure of the droplet. The activation





potential increases with increase of the droplet diameter or decrease of surface tension $\sigma_{s/a}$ and the
$\sigma_{s/a}$ value is sensitive to the organic surfactant effect. The two important parameters, the water activity
($a_w$) and surface tension ($\sigma_{s/a}$), are dependent on the composition of the aerosol particles, assuming
those particles have the same properties as their corresponding bulk solutions. The effects of organics
on the CCN activity have been extensively investigated; however, many outstanding questions still
remain. Sorjamaa et al. (2004) suggested that the partitioning of surfactant had to be considered when
evaluating the Kelvin effect and the Raoult effect. According to their experimental results, the
surfactant partitioning could alter the Raoult effect so that the change is large enough to depress CCN
activity. However, another experiment conducted by Engelhart et al. (2008) revealed that the organics
in aged monoterpene aerosols could depress surface tension by about 0.01 N m$^{-1}$ and hence increase
CCN activity. Ovadnevaite et al. (2017) also presented observational and theoretical evidences that the
decrease of surface tension could prevail over the Raoult effect, which led to the increase of CCN
activity. Salma et al. (2006) isolated humic-like substances (HULIS) from PM2.5 fraction aerosol
samples and investigated the surface tension properties of the HULIS pure solutions. Their results
showed that thermodynamic equilibrium on surface could only be reached after several hours. Because
the depression of surface tension was controlled by diffusion of surfactants from the bulk of the droplet
to its surface, the extent of the actual decrease of surface tension was hence kinetically limited. A
hybrid model proposed by Petters and Kreidenweis (2013) was used to predict the effects of surfactants
on the CCN activity. The model predicted strong effects of the surfactants on ternary systems where
common ions were present. However, due to the limited measurement techniques, the available
laboratory data were still not sufficient to support this prediction and more solid data were needed to



validate the surfactant effects on the CCN activity.
The CCN activity can be characterized by the hygroscopicity parameter κ that was initially proposed by
Petters and Kreidenweis (2007). Aerosol hygroscopicity represents the ability of the particles to grow
by absorbing water vapor from the atmosphere and the extent to which the particles are hygroscopic
can be evaluated by the κ values, which can be determined from the H-TDMA or CCNc measurements.
The κ values were measured worldwide extensively either in the field measurements or in the
laboratory experiments and depending on the organic content of the particles, a wide range of κ values
were reported in the literatures. Cerully et al. (2011) showed that the κ values measured in 2007 by
Flow-Streamwise Thermal Gradient CCN Chamber (CFSTGC) ranged mostly between 0.1 and 0.4 in a
forest environment in Finland. Hong et al. (2014) obtained the average κ values of 0.15 (110 nm) and
0.28 (102 nm) measured by H-TDMA at the same site in 2010. Chang et al. (2010) used an AMS to
measure aerosol chemical composition and a mole ratio of atomic oxygen to atomic carbon (O/C) at a
rural site in Canada. They reported a relationship between the κ values of organics and the O/C ratio as
$\kappa_{org}=(0.29\pm0.05)*(O/C)$. Tritscher et al. (2011) conducted smog chamber experiments for
measurements of the κ values of aging secondary organic aerosols and they found that the κ was a
sensitive indicator of the SOA properties.
Although the κ values were reported under different environments in many locations, only a few
studies were conducted to measure κ in the Pearl River Delta (PRD) region (Cheung et al. 2015;
Schurman er al. 2017). Jiang et al. (2016) compared the κ values between wintertime (0.18-0.22) and



summertime (0.17-0.21) in Guangzhou. Cai el al. (2017) reported the $\kappa$ values of about 0.4-0.6 and
0.2-0.3 measured by the H-TDMA respectively in Cape Hedo (Japan) and in Guangzhou (China).
Alternatively, the average $\kappa$ values can be predicted by the ZSR mixing rule (Zdanovskii, 1948; Stokes
and Robinson, 1966) which is based on the chemical composition of the aerosol particles from the
AMS measurement. Liu et al. (2014) reported the $\kappa$ values of 0.22 to 0.32 using the ZSR mixing rule,
consistent with the values (0.25 to 0.34) based on the H-TDMA measurement.
Once the $\kappa$ values were determined, they could then be employed to predict the CCN activity that was
characterized by three important parameters: activation diameter ($D_{50}$), CCN number concentration
($N_{CCN}$), and activation ratio (AR). Until now, the CCN activity (thus the above three parameters) can be
determined using the following three methods:
(1) The combination of Cloud Condensation Nuclei counter (CCNc) and Scanning Mobility Particle
Sizer (SMPS). The CCN number was measured by the CCNc at different supersaturation ratios (SS,
typically 0.05% ~ 1%). Meanwhile, the $D_{50}$ and size-resolved activation ratios could be measured by
combining the CCNc with a differential mobility analyzer (DMA) and a condensation particle counter
(CPC)(Moore et al., 2011; Deng et al., 2011), referred to as Scanning Mobility CCN Analysis (SMCA)
based on measurements from a SMPS (DMA+CPC) and a CCNc. This method can measure the
size-resolved CCN distributions at a high time resolution (Moore et al., 2010) and has been applied in
laboratory experiments (Asaawuku et al., 2009) and filed campaign (Moore et al., 2008) to measure
CCN activity.
(2) The ZSR method based on chemical composition measurements. The CCN concentrations were



inverted from the chemical composition and the size distribution of the aerosol particles measured
respectively from the aerosol mass spectrometer (AMS) and SMPS (Moore et al., 2012; Meng et al.,
2014). The κ was then calculated from the ZSR mixing rule. In general, the particles were assumed to
be internally mixed, which might lead to a large uncertainty (up to 80%) in predicting $N_{CCN}$ in some
cases (Wang et al., 2010).
(3) The H-TDMA method. The size-resolved CCN distribution and activation ratios could be
determined from hygroscopocity and size distribution measured using the H-TDMA (Good et al., 2010;
Wu et al., 2013). The H-TDMA measured the distribution of hygroscopic growth factor (Gf) at a fixed
relative humidity for a selected diameter of aerosol particles. Väisänen et al. (2016) reported that the
measured $N_{CCN}$ with the H-TDMA agreed well with that from in-cloud prediction, where the sample
was collected from a tower approximately 224 m above the surrounding lake level. On the other hand,
Chan (2008) attributed differences in κ from H-TDMA and CCN measurements to sparingly soluble
organics that did not easily deliquesce in the former measurements.
The PRD region is one of the most economically invigorating regions in China. This region is
subjected to severe air pollution due to intense human activities and insufficient pollution control
measures. High particle loading leads to both visibility degradation and large cooling effects due to
decrease of solar radiation. During wintertime, high concentrations of fine particles also cause severe
haze events that pose health risk on human beings at the regional scale. It is hence an ideal location to
investigate the influence of local anthropogenic emissions on the particles properties. However, there is
still lack of understanding on the relationship between the CCN activity and its controlling factors (e.g.,



chemical composition and hygroscopicity of aerosol particles), hindering policy-makers to propose
effective measures for air pollution control.
In this study, we used the SMCA, H-TDMA, and HR-ToF-AMS to respectively measure CCN activity,
hygroscopicity and chemical composition. We reported the relationship between CCN activity and
hygroscopicity/chemical composition of aerosol particles in the PRD region, where only a few studies
on such relationship were available in the literature. The measurements were performed during
wintertime 2014 (November and December). The CCN properties were predicted based on the
combined SMCA, H-TMDA and HR-ToF-AMS measurements. The methods employed to predict the
CCN concentrations were evaluated and the impact of organics on CCN concentrations was discussed.
**2    Experiments and data analysis**
**2.1 Measurement site**
The field measurements were conducted at the Chinese Meteorological Administration (CMA)
Atmospheric Watch Network (CAWNET) Station in Panyu, Guangzhou, China, during wintertime
2014 (November and December). The Panyu Station is located at the center of the PRD region and at
the top of Dazhengang Mountain (23°00'N, 113°21'E) with an altitude of about 150 m. No significant
local emission sources were around the site. Detailed description of the measurement site and
instruments (i.e., the H-TDMA and the AMS) can be found elsewhere (Cai et al., 2017; Qin et al.

19    2017).

**2.2 Instrumentation**
**2.2.1    Aerosol hygroscopicity measurements**



Size-resolved aerosol hygroscopicity and particle number size distribution (PNSD) were measured by a
H-TDMA which was developed by Tan et al. (2013). The hygroscopicity data were only available in
November due to the failure of the H-TDMA during December. An aerosol sampling port equipped
with a $PM_{1.0}$ impactor inlet was used during the measurement period. Ambient sampling flow first
passed through a Nafion dryer (Model PD-70T-24ss, Perma Pure Inc., USA) to achieve a RH of <10%.
We considered the particles to be dry when the RH values were less than 10%. The particles were
subsequently charged by a neutralizer (Kr85, TSI Inc.) and size-selected by a differential mobility
analyzer (DMA1, Model 3081L, TSI Inc.). The mono-disperse particles with a specific diameter ($D_0$)
were then introduced into a Nafion humidifier (Model PD-70T-24ss, Perma Pure Inc., USA) under a
fixed RH of (90 ±0.44) %. Another differential mobility analyzer (DMA2, Model 3081L, TSI Inc.) and
a condensation particle counter (CPC, Model 3772, TSI Inc.) were used to measure the number size
distribution of the humidified particles (Dp). Thus, growth factor (Gf) of the particles can be
calculated:
$$Gf = \frac{Dp}{D_0}$$  (2)
During the campaign, we selected five dry mobility diameters (40, 80, 110, 150, and 200 nm) for the
H-TDMA measurements. The measurements were performed continuously except for regular
calibration of the instrument. We used standard polystyrene latex spheres and ammonium sulfate to
perform the DMA calibration to ensure the instrument to function normally.
**2.2.2 Size-resolved CCN activity measurements**
Size-resolved CCN spectra and activation ratios were measured with the SMCA initially proposed by



Moore et al. (2010)). In this work, the SMCA consisted of a CCNc-100 (DMT Inc.), a differential
mobility analyzer (DMA, Model 3081L, TSI Inc.) and a condensation particle counter (CPC, Model
3787, TSI Inc.). In the SMCA system, the combined DMA and CPC were used as a scanning mobility
particle sizer (SMPS) during the measurements. The dry particles after the Nafion dryer were
neutralized by the Kr85 neutralizer and were subsequently classified by the DMA. The mono-disperse
particles were split into two streams: one to the CPC for measurement of total particle number
concentration ($N_{CN}$) and another to the CCNc-100 for measurement of the CCN number concentration.
The aerosol and CPC flow rate was both 1.0 L min$^{-1}$ for the DMA and the CPC (0.5 L min$^{-1}$ makeup
flow and 0.5 L min$^{-1}$ sample flow), respectively. The CCNc-100 drew another aerosol flow rate of 0.5 L
min$^{-1}$. The SMCA was protocoled to measure particles at a mobility diameter range of 10 - 400 nm. The
supersaturation in the CCNc-100 was set to be 0.1%, 0.2%, 0.4% and 0.7% respectively for each
measurement cycle. The CCNc-100 was regularly calibrated with ammonium sulfate particles at the
four SS (0.1%, 0.2%, 0.4%, and 0.7%). Similarly, the DMA was calibrated with standard polystyrene
latex spheres before and after the campaign for quality assurance and control.
**2.2.3 Aerosol chemical composition measurements**
An Aerodyne high-resolution time-of-flight aerosol mass spectrometer (HR-ToF-AMS) was employed
in the campaign to measure non-refractory PM$_1$ chemical composition (bulk and size-resolved)
including sulfate, nitrate, ammonium, chloride and organics. The refractory components such as black
carbon, sea salts and crustal species cannot be measured by this instrument. Detailed description of the
HR-ToF-AMS can be found elsewhere (DeCarlo et al., 2006; Jimenez et al., 2003). Here only a brief
description relevant to the measurements is given. The instrument was operated in three modes (pToF,





V, and W mode). Particle size distribution could be obtained based on time-of-flight of the particles in
pToF mode. In V and W modes, the resolving power of the mass spectrometer was approximately 2000
and 4000, respectively. The instrument collected alternatively 5-min average mass spectra for the V +
pToF modes and the W mode. The monodisperse pure ammonium nitrate ($NH_4NO_3$) particles selected
by a DMA (400 nm) were used weekly in the ionization efficiency (IE) calibration. Background signals
were obtained daily for about 30 minutes by introducing filtered ambient air with a HEPA filter in the
sample flow. Before and after the measurement, the sampling flow rate was calibration with a Gilian
gilibrator. We also generated PSL (Duke Scientific) and ammonium nitrate particles in a size range of
178~800 nm to calibrate the pToF size. Note that the mass concentrations were too low for particle
diameters smaller than 65 nm and data for those particles were hence discarded in this study. A more
detailed description of the AMS performance during the measurements can be found in Qin et al. (2017)
and Cai et al. (2017).
The AMS measured size-resolved chemical composition of particles in vacuum aerodynamic diameter
($D_{va}$). It is hence necessary to convert aerodynamic diameter to mobility diameter in order to compare
the AMS data and the SMCA data. We adopted the equation derived by DeCarlo et al. (2004) to do the
conversion. Here we assume a density of 1700 kg m$^{-3}$ for particles measured by the AMS (DeCarlo et
al., 2004)
**2.3 Data processing and methodology**
**2.3.1 Hygroscopicity**



Due to the effects of diffusing transfer function, the measured distribution function (MDF) given by
H-TDMA is only a skewed and smoothed integral transform of the actual growth factor probability
density function (Gf-PDF) of the particles (Gysel et al., 2009). Here the TDMAfit algorithm
(Stolzenburg and McMurry, 2008) was applied to narrow the uncertainties caused by the diffusion
broadening. The TMDAfit algorithm describes the Gf-PDF as a combination of several (usually smaller
than three) lognormal distribution functions, in which the parameters of each mode are considered as
mean Gf, standard deviation, and number fraction. The detailed data inversion process of the H-TDMA
instrument can be found in Tan et al. (2013).
As mentioned in the introduction, the CCN activity can be represented by a widely used hygroscopicity
parameter κ (Petters and Kreidenweis, 2007). According to the κ-Köhler theory, for a known
temperature, κ and Gf can be related via eq. 3 (Petters and Kreidenweis, 2007):
$$\kappa = (Gf^3 - 1)\left[\frac{1}{RH}\exp\left(\frac{4\sigma_{s/a}M_w}{RTGf\rho_w D}\right) - 1\right] \qquad (3)$$
where $\rho_w$ is the density of water (about 998.34 kg m$^{-3}$ at 293K), $M_w$ is the molecular weight of water
(0.018 kg mol$^{-1}$), $\sigma_{s/a}$ is the surface tension of the solution/air interface and here pure water is
tentatively assumed for the solution ($\sigma_{s/a}$ =0.0728 N m$^{-1}$ at 293K), R is the universal gas constant (about
8.31 J mol$^{-1}$K$^{-1}$), T is thermodynamic temperature in Kelvin, and D is the particle diameter (in meter).
**2.3.2 CCN activation**
The $N_{CN}$ and $N_{CCN}$ data were respectively measured by the SMPS and the CCNc-100 and they were
used to calculate the size-resolved CCN activation ratios (AR) which was defined as the ratio of $N_{CCN}$
to $N_{CN}$ at each particle size. The activation ratio can be obtained by fitting the ratio with the sigmoidal



function with respect to Dp:
$$\frac{N_{CCN}}{N_{CN}} = \frac{B}{1+(\frac{D_p}{D_{50}})^c}$$    (4)
where Dp is the particle dry diameter, B, $D_{50}$ and c are fitting coefficients that represent the asymptote,
the slope, and the inflection point of the sigmoid, respectively (Moore et al., 2010). $D_{50}$ is also called
the critical diameter or the activation diameter, that is, the diameter at which 50% of the particles are
activated at a specific SS.
Alternatively, the hygroscopicity parameter κ can be calculated from the critical saturation ratio (Sc)
and $D_{50}$ from the following equation (Petters and Kreidenweis, 2007):
$$\kappa = \frac{4A^3}{27D_{50}^3(\ln Sc)^2} \; , \quad A = \frac{4\sigma_{s/a}M_w}{RT\rho_w}$$    (5)
**2.3.4 CCN prediction based on H-TDMA and AMS measurements**
The $N_{CCN}$ can be predicted based on either the aerosol hygroscopicity data (measured by the H-TDMA)
or the AMS data. Figure 1 is the schematic diagram we followed to predict $N_{CCN}$ based on the above
two measured datasets. In the first approach, we assumed the critical hygroscopicity parameter $\kappa_{critical}$ to
be a function of the particle diameter and the supersaturation ratio (denoted as $\kappa_{critical}$(Dp, SS)). The
$\kappa_{critical}$ was hence defined as the point at which all the particles were activated at a specific diameter and
a specific SS. Here we measured hygroscopicity using the H-TDMA at five dry diameters and the CCN
concentrations at four SS. We calculated the $\kappa_{critical}$(Dp, SS) using eq.5 for a known diameter and SS.
The particle with a κ value higher than $\kappa_{critical}$(Dp, SS) was considered to be activated as CCN. The
activation ratio for a specific diameter at a specific SS was obtained by integrating the κ-PDF for κ >
$\kappa_{critical}$(Dp, SS). The size-resolved activation ratio ($AR_{SR}$) was determined by fitting the AR(Dp, SS)





using the eq.4. Thus, the calculated $N_{CCN}$ can be expressed as:
$$N_{CCN}(SS) = \int_0^\infty AR_{SR}(Dp, SS)N_{CN}(Dp)dDp \qquad (6)$$
In the second approach, we calculated the κ value according to the ZRS rule based on the AMS
measurements:
$$\kappa = \sum_i \varepsilon_i \kappa_i \qquad (7)$$
where $\varepsilon_i$ is the volume fraction of each component in the particles, $\kappa_i$ is the κ value of each component.
The AMS only provided the ion concentrations during the measurements, while the ZSR rule required
the volume fraction and hygroscopicity of each component. A simplified ion pairing scheme developed
by Gysel et al. (2007) was used to reconstruct the $NH_4^+$, $SO_4^{2-}$ and $NO_3^-$ measured by the AMS:

$$n_{NH_4NO_3} = n_{NO_3^-}$$

$$n_{H_2SO_4} = \max(0, N_{SO_4^{2-}} - n_{NH_4^+} + n_{NO_3^-})$$

$$n_{NH_4HSO_4} = \min\left(2n_{SO_4^{2-}} - n_{NH_4^+} + n_{NO_3^-}, n_{NH_4^+} - n_{NO_3^-}\right)$$

$$n_{(NH_4)_2SO_4} = \max\left(n_{NH_4^+} - n_{NO_3^-} - n_{SO_4^{2-}}, 0\right)$$

$$n_{HNO_3} = 0, \qquad (8)$$
where n denotes the number of moles of each component (i.e., $NH_4^+$, $SO_4^{2-}$ and $NO_3^-$ ). Here we used
the ADDEM proposed by Topping et al.(2005) to calculate the κ values of the inorganic species and
those of the organics were tentatively assumed to be 0.1 (Meng et al., 2014). Table 1 lists the κ values
of the relevant species used in the study based on the calculations and the above assumption.



The $D_{50}$ can be calculated from the above κ values using eq.4. The CCN concentration is obtained by
integrating the cloud nuclei concentration for particles larger than $D_{50}$ based on the particle size
distribution:
$N_{CCN}(SS) = \int_{D_{50}}^{\infty} N_{CN}(Dp)dDp$                                      (9)
**3. Results and discussion**
**3.1 Overview**
Table 2 summarizes the observed CCN activity during the campaign. Overall, the average $N_{CCN}$ at 0.1,
0.2, 0.4, and 0.7% SS were about 3100, 5100, 6500, and 7900 cm$^{-3}$, respectively. The average
activation ratios (AR) at the above four SS were 0.26, 0.41, 0.53 and 0.64, respectively. The average
$D_{50}$ at the above four SS were 156, 107, 78 and 58 nm, respectively. The $N_{CCN}$ at 0.7% SS was
respectively lower than those of the previous measurements (10731 cm$^{-3}$ at 0.67% SS) in July 2006 in
Guangzhou (Rose et al., 2010), but much higher than those measured (2085 cm$^{-3}$ at 0.7% SS) in May
2011 in Hong Kong (Meng et al., 2014), while the AR was lower than those from the previous
measurements in Guangzhou (0.59 at 0.67% SS, Rose et al., 2010) and similar to those from the
measurements in Hong Kong (0.64 at 0.7% SS, Meng et al., 2014). The $D_{50}$ was larger than that in the
previous measurements in Guangzhou (49 nm) and in Hong Kong (47 nm), due to the lower particle
hygroscopicity in Guangzhou. The differences of the $\kappa_{CCN}$ values between the two measurements (0.21
in this winter campaign vs 0.28 during the summer season in Guangzhou both at 0.7% SS) suggested
that the particles in the summer were in general more hygroscopic and hence were more readily
activated than those in the winter, implying different chemical composition of the particles between the



two distinct seasons.
Figure 2 shows the average mass fraction of NR-PM$_1$ bulk composition and size-resolved (64-731 nm)
composition. The organics was dominant in the bulk NR-PM$_1$ (50%), followed by sulfate (26%) and
nitrate (12%) (Fig.2a). The mass fraction of the organics decreased with the size (Fig. 2b), from 69% at
64 nm to 42% at 397 nm. The mass fraction of organics at 397 nm was close to that of NR-PM$_1$ bulk,
due to the fact that the PM$_1$ mass is dominated by particles in a diameter range of 200~500 nm (Tan et
al., 2016). Aerosol particles with larger sizes were more readily exposed to complex atmospheric
composition during their aging process, contributing from partitioning between gas phase and particle
phase via photochemical reactions, surface heterogeneous reactions etc. In comparison, the dominant
NR-PM$_1$ species observed in Hong Kong were sulfate (51.0%) and organics (28.2%) (Lee et al., 2013),
significantly different from our measurements, due probably to different origins of the dominant air
masses between the two seasons. The measurement site in Guangzhou was impacted predominantly by
the air mass from north, where straw burning contributes to a high mass fraction of organics matter
(Cao et al., 2008).
Figure 3 shows the κ values based respectively on the CCN ($\kappa_{CCN}$), AMS ($\kappa_{AMS}$), and H-TDMA
($\kappa_{H-TDMA}$) measurements, along with the measured particle number size distribution (PNSD, 10-400 nm)
during the campaign. The shadow area represents the interquartile range of the PNSD. A distinct peak
at around 90 nm was observed from the PNSD (Fig. 3). The $\kappa_{AMS}$ was calculated based on the
size-resolved chemical composition, assuming the particles are internally mixed. At 0.7% SS, the D$_{50}$





was about 58 nm. Hence no $\kappa_{AMS}$ was reported at this SS since we only measured particle composition
above 63 nm using the AMS in this study. The κ values were shown in the interquartile range, with the
largest variation from the CCN measurements (Fig. 3). Figure 3 showed that the $\kappa_{H\text{-}TDMA}$ values were
lower than those of the corresponding $\kappa_{CCN}$ at most of the SS, consistent with the previous observation
(Pajunoja et al., 2015). This was probably due to the facts that particles contain a certain fraction of low
solubility composition, such as secondary organic aerosols (SOA), contributing differently to
hygroscopic growth and CCN activation. The available AMS data (Fig. 3) show that the $\kappa_{AMS}$ values
were lower than the corresponding $\kappa_{CCN}$ and $\kappa_{H\text{-}TDMA}$ values at all size ranges and the differences
become larger with increasing particle sizes. This was probably due to underestimated hygroscopicity
in the organic composition when using the AMS data, since we assumed a κ value of 0.1 for all
organics at all particle sizes. The hygroscopicity increased with particle diameters due to aerosol aging
which increased the hygroscopic organic contents. The measured $\kappa_{mean}$ values fall in a range of
0.22-0.30 for the particle sizes of 40-200 nm measured by H-TDMA in this study. The other aerosol
hygroscopicity measurement in PRD (Jiang et al., 2016) reported the $\kappa_{mean}$ values ranging from 0.18 to
0.22 in 2012 winter season and 0.17 to 0.21 in 2013 summer season, suggesting an increase of the
aerosol hygroscopicity which might result from an increasing mass fraction of nitrate in recent years
(Zhang et al., 2015; Itahashi et al., 2018).
Figure 4 shows the activation ratios (AR) measured by SMCA at four supersaturation ratios (0.1, 0.2,
0.4, 0.7%) for particles below 300 nm. The activation curves obtained in this study were segmented
into three sections: a steady rise at low ARs, a middle sharp increase, and a plateau at almost 100% AR.



We defined the steepness as the rate at which the AR increased with the particle sizes. Figure 4 shows
the steepness increased with the SS, indicating that the curves became steeper with the SS and a larger
variation of the $D_{50}$ was expected. In addition, the CCN activity was more sensitive to particle
diameters at higher SS, which can be seen from partial derivative of $\kappa_{critical}$ by $\partial D_{50}$ (eq. 5):
$\frac{\partial \kappa_{critical}}{\partial D_{50}} = -\frac{4A^3}{9D_{50}^4 (\ln Sc)^2}$ (10)
For a certain SS, the $\kappa_{critical}$ value became more sensitive to $D_{50}$ with decrease of the $D_{50}$.
Meanwhile, a high SS usually led to a low $D_{50}$. Therefore, the AR would vary with Dp more readily at
higher SS and the curve would become steeper. A higher SS allowed a smaller particle to be activated
and the activation curve became steeper, and vice versa for a lower SS.
The steepness of activation curve was also associated with the heterogeneity of aerosol chemical
composition, that was, a steeper activation curve meant that aerosol particles had higher similarity in
hygroscopicity. A bimodal distribution (peaks at about 1-1.1, and 1.5-1.7 Gf) of the Gf-PDFs was
observed along the Gf coordinate at all the five sizes of the particles measured by H-TDMA in this
study (Fig. 5), corresponding respectively to the less- and more-hygroscopic modes. The peak in the
less-hygroscopic mode declined and was shifted to smaller Gf, while the one in the more-hygroscopic
mode climbed and shifted to larger Gf with increase of the diameter of the particle, indicating larger
particles tend to be internally mixed. Here a parameter σ is introduced to illustrate the deviation of
Gf-PDF (Gysel et al., 2009):
$Gf_{mean} = \int_0^\infty Gf c(Gf) dGf$ (11-1)



$\sigma = (\int_0^\infty (Gf - Gf_{mean})^2 c(Gf) dGf)^{\frac{1}{2}}$                    (11-2)
where the $c(Gf)$ denotes Gf-PDF and $Gf_{mean}$ denotes number weighted mean $Gf$. The σ was
employed as a measure of the spread of Gf-PDF which represents the heterogeneity of aerosol chemical
composition (Sjogren et al., 2008; Liu et al., 2011). A small σ indicated that the heterogeneity of
aerosol chemical composition was low and aerosol particles had higher similarity in hygroscopicity.
The parameter $C$ determined the shape of activation curve which was segmented into steep and
smooth parts. A small $C$ value means a steep activation curve and vice versa. Here an activation curve
was assumed to be steep when the $C$ values are lower than the lower quartile of all the $C$ values,
while the activation curve was considered to be smooth when the $C$ values are higher than the upper
quartile of all the $C$ values. Table 3 summarizes the σ values of GF-PDF for the corresponding steep
and smooth activation curve at the four supersaturations. In general, the σ increased with the diameter,
indicating that larger particles had higher heterogenerity of aerosol chemical composition. Meanwhile,
the σ values for smooth curve were generally higher than the σ values for steep curve. The results
implied that the shapes of activation curves were related to the heterogeneity of aerosol chemical
composition.
**3.2 Impact of organics on CCN activity**
Figure 6 shows the relationship between the $D_{50}$ obtained from the SMCA measurements and the
size-resolved mass fractions of organics ($f_{org}$) at three supersaturation ratios (0.1%, 0.2%,and 0.4% SS).
In general, the $D_{50}$ increased with $f_{org}$ at the three SS, with a slope of 127, 667, 21, and a fitting
coefficient ($R^2$) of 0.47, 0.31, 0.1 at 0.1%, 0.2%, and 0.4% SS, respectively. The particles usually
became less hygroscopic with increase of the organic fractions ($f_{org}$), which then required larger



particles to be activated. At lower SS, better correlations were found between the $f_{org}$ and the $D_{50}$
because the $D_{50}$ was more sensitive to hygroscopicity (The activation ratios increase more slowly with
particle sizes at lower SS as shown in Fig. 4). It was hence more obvious at lower SS that the
modification of the particle hygroscopicity caused by the change of the mass fraction of organics
matter could greatly modify the $D_{50}$ which might further affect the CCN activity. At higher SS,
according to eq. 5, particles were more easily activated as CCN and the change of particles
hygroscopicity would not significantly alter the CCN activity.
Organics can affect the CCN activity via two opposite ways: they can decrease the CCN activity by
increasing the less hygroscopic organic fraction of the particles and thus increase the $D_{50}$ as shown in
Fig. 6; they can also increase the CCN activity by decreasing the surface tension of the particles. The
latter effect has been demonstrated experimentally. For example, an increase of CCN activity was
observed when organics were added to sulfate ammonium (Engelhart et al., 2008). In this study, we
investigated the impacts of organics on CCN activity through adjusting the value of surface tension
until the calculated AR values based on H-TDMA measurements agree with those obtained from
SMCA measurements (measured AR). As shown in Fig. 7, the calculated AR values were
systematically lower than the corresponding measured ones if the surface tension of bulk pure water
(0.072 N m$^{-1}$) was assumed when calculating the AR from the H-TDMA measurements.
The surface tension of a nanoparticle was substantially different from that of its bulk solution due to the
curvature effect. The effects of size and composition on the surface tension were currently not well





understood. Here we proposed an approach to evaluate the impact of organics on the surface tension
($\sigma_{s/a}$) based on the fraction change of the calculated AR to the measured AR. We defined this fraction
change ($\delta_{AR}$) as a function of surface tension, diameter, and supersaturation:
$\delta_{AR}(\sigma_{s/a}, Dp, SS) = \frac{AR_m(Dp,SS) - AR_c(\sigma_{s/a}, Dp, SS)}{AR_m(Dp,SS)} \times 100\%$  (12)
where $AR_m(Dp, SS)$ is the measured AR for a certain diameter and SS, $AR_c(\sigma_{s/a}, Dp, SS)$ is the
calculated AR for a certain diameter, SS, and $\sigma_{s/a}$. We excluded particles at the size of 200 nm because
they were easily activated even at 0.1% SS and the $\delta_{AR}$ was expected to be independent of $\sigma_{s/a}$. Here the
$\sigma_{s/a}$ value varied between 0.03 and 0.072 N m$^{-1}$ (surface tension of pure water). Figure 8 shows the $\delta_{AR}$
as a function of $\sigma_{s/a}$ for the four particle diameters (40, 80, 110, 150 nm). The $\delta_{AR}$ decreased with
increase of the $\sigma_{s/a}$ for all given particle sizes, changing more rapidly for smaller particles (i.e., from
200% to -100% for 40 nm) than bigger particles (i.e., from 20% to -10% for 150 nm). The $R^2$ between
measured AR and predicted AR for a certain diameter and four supersaturations at $\sigma_{s/a}$=0.072 N m$^{-1}$
were 0.35, 0.93, 0.95 and 0.91, respectively. The $\delta_{AR}$ values reached zero when the $\sigma_{s/a}$ was set to be
about 0.054 N m$^{-1}$ for 40, 80, and 110 nm particles, and 0.062 N m$^{-1}$ for 150 nm particles, with a $R^2$ of
0.88, 0.94, 0.94 and 0.88 respectively. As a compromise, here we adopt a $\sigma_{s/a}$ value of 0.058 N m$^{-1}$
(denoted as $\sigma_{s/a}^{*}$) to predict AR. This $\sigma_{s/a}^{*}$ value increased significantly the $R^2$ compared to that based
on pure water assumption (0.072 N m$^{-1}$) for 40 nm particles, while it was reasonable well for other
sizes of particles (80, 110, 150 nm). The AR was then recalculated using the $\sigma_{s/a}^{*}$ value and the
prediction was significantly improved (Fig. 9). The results demonstrated that partitioning of organics
into aerosol particles would decrease their surface tension. Therefore, the pure water assumption for
surface tension would lead to high uncertainties when it applied to predict the activation ratios of the



aerosol particles at a certain size. Note that we did not consider the effects of individual organics due to
the limited data from the chemical composition measurements. How chemical composition affects the
surface tension of the particles is yet to be investigated.
**3.3 The $N_{CCN}$ prediction**
**3.3.1 The $N_{CCN}$ prediction based on the H-TDMA measurements**
In this study, we used several approaches to predict the $N_{CCN}$ based on the H-TDMA measurements,
from either the activation curve or the $D_{50}$. Table 4 summarizes the methods that were used to predict
the $N_{CCN}$, along with the slope and $R^2$ between the predicted and the measured values. The mixing state
of the aerosol particles is an important parameter in determining the $N_{CCN}$. The activation curve
represented actual mixing state, while the $D_{50}$ approach assumed that all particles were internally mixed.
Scheme 5 in Table 4 was the method based on the activation curve with the new $\sigma_{s/a}^*$ (0.058 N m$^{-1}$). Eq.
6 and Eq. 9 were respectively used to calculate the $N_{CCN}$ following scheme 1, 2, 5, and the rest of the
schemes. Scheme 5 (real time activation curve using $\sigma_{s/a}^*$) provided the best $N_{CCN}$ predicted value
(closest to the measured one), followed by scheme 3 (real time $D_{50}$) > scheme 4 (average $D_{50}$) >
scheme 1 (real time activation curve) > scheme 2 (average activation curve). The $R^2$ values for all the
approaches were in general high (around 0.93). The CCN prediction based on scheme 2 led to the
largest underestimation over the measured values. In general, the real time data (scheme 1 and 3) gave
better predicted $N_{CCN}$ than the corresponding average data (schemes 2 and 4).
Figure 10 shows the correlation between the measured $N_{CCN}$ and the predicted $N_{CCN}$ from scheme 1-5
at the four SS. For scheme 1-4, the predicted $N_{CCN}$ values were found to be the largest deviation from





the corresponding measured ones at 0.1% SS among all the approaches, probably due to the pure water
assumption for surface tension ($\sigma_{s/a}$ = 0.072N m$^{-1}$). Meanwhile, because the CCN activity was sensitive
to hygroscopicity of the particles at low SS, the uncertainties of hygroscopicity data would lead to large
errors in the prediction of CCN. As discussed in the previous section, the $D_{50}$ was more sensitive to the
$\sigma_{s/a}$ at lower supersaturations, leading to a large deviation of the $N_{CCN}$ from the measured value. The
best agreement between the calculated AR and the measured AR was seen using scheme 5 as the slopes
at the four SS were close to 1(Fig. 10, q-t).
**3.3.2 The $N_{CCN}$ prediction based on AMS measurements**
We proposed five approaches based on H-TDMA measurements to predict the $N_{CCN}$ in the previous
section. Alternatively, we can calculate the $N_{CCN}$ based on AMS measurements. Here we proposed four
methods based on either size-resolved chemical composition or bulk PM$_1$ chemical composition from
the AMS measurements (Table 5). Here we assumed that the particles were internally mixed and the
median $\kappa_{AMS}$ obtained from bulk composition was 0.28, higher than those from size-resolved
composition (0.24-0.26 in Fig. 3), probably due to a higher mass fraction of inorganic matters in bulk
NR-PM$_1$ (Fig. 2). We excluded the size-resolved data at 0.7% SS due to their poor quality. Figure 11
shows the correlation between the measured and predicted $N_{CCN}$ from schemes 6-9. The $N_{CCN}$ was
under-predicted at 0.1% SS and was over-predicted at 0.7% SS. We proposed three potential factors
that might impact $N_{CCN}$ prediction based on AMS measurements. (1) The assumed $\kappa_{org}$ values were
probably underestimated for particles larger than 100 nm, leading to the underestimated $N_{CCN}$ at low SS.
As shown in Fig. 3, the predicted $\kappa$ shows a larger deviation from the measured value for a larger





particle. The $D_{50}$ values were more sensitive to particle hygroscopicity at lower SS as discussed in the
previous section. (2) The pure water assumption for surface tension. As we have shown in the previous
section, the $\sigma_{s/a}$ values for the aerosol particles were found to be much smaller than the $\sigma_{s/a}$ for pure
water (0.072 N m$^{-1}$). As a result, the pure water assumption for surface tension led to the $N_{CCN}$
underestimation. In addition, again the $D_{50}$ was more sensitive to $\sigma_{s/a}$ at the low SS. (3) The exclusion
of black carbon (BC) particles and the mixing state assumption. The BC particles were known to be
non-hygroscopic and had a low CCN activity. During the campaign period, the average BC
concentration was about 5.91 μg/m$^3$ which accounts for 7 % in PM$_{2.5}$. The assumption of no BC
particles would lead to the overestimation of $N_{CCN}$. The assumption of particles being internally mixed
in the AMS measurements would lead to an overestimation of the $N_{CCN}$ when the ambient particles tend
to be externally mixed (Wang et al., 2010; Sánchez Gácita et al., 2017). However, the internal mixing
assumption seems to play a minor role in predicting the $N_{CCN}$ at 0.1% SS since the particles at about
140-180 nm tend to be internally mixed as shown in Fig. 5. In this case, the $\kappa_{org}$ assumption and the
pure water assumption played more important roles than the mixing state assumption at low SS (i.e.,
0.1% SS). Figure 11 shows significant $N_{CCN}$ underestimation at 0.1% SS (panels a, e, i, m), while more
or less comparable to the measured $N_{CCN}$ at higher SS (i.e. 0.2%, 0.4%, 0.7%). The difference between
the $\kappa_{AMS}$ and $\kappa_{CCN}$ became smaller and the corresponding $D_{50}$ value decreased with the increase of the
SS so that the impacts of the $\kappa_{org}$ assumption and the pure water assumption became minor with the
increase of the SS. Instead, the internal mixing state assumption would play a more important role in
the prediction (Meng et al. 2014). As shown in Fig. 5, the peak height and area of the less-hygroscopic
mode became larger for the smaller size particles (i.e, 40 nm particles), implying that small particles



were likely to be externally mixed, that is, the non or less hygroscopic species including BC and
insoluble organics were less likely coated with inorganics salts. Hence the internal mixing assumption
could lead to an overestimated $N_{CCN}$.
As discussed above, the two important parameters ($\kappa_{org}$ and $\sigma_{s/a}$) had significant impacts on the $N_{CCN}$
prediction. We denoted $\kappa_{org}^*$ and $\sigma_{s/a}^*$ as important representations respectively for hygroscopicity and
surface tension contributed from organics. We also pointed out that the $\kappa_{org}$ was dependent on the
particle size and hence here we further assumed the $\kappa_{org}^*$ values to be 0.15 and 0.1 respectively for
particles larger and smaller than 100 nm. Note that we previously assumed the $\kappa_{org}$ to be 0.1 for all
particle sizes. Here we gave an example of the improvements at 0.1% SS when the $\kappa_{org}$ and $\sigma_{s/a}$ values
were respectively replaced with the $\kappa_{org}^*$ and $\sigma_{s/a}^*$ ones (Fig. 12). The $\kappa_{AMS}$ value calculated at 0.1% SS
based on $\kappa_{org}^*$ was 0.288, very close to the corresponding $\kappa_{CCN}$ value (0.30), indicating that an
improvement was made for the $N_{CCN}$ prediction when including the $\kappa_{org}^*$ value. The $N_{CCN}$ prediction
could be greatly improved when include both $\sigma_{s/a}^*$ and $\kappa_{org}^*$ in the calculation (i.e, from 44% in Fig. 11a
to 4% in Fig. 12b). In addition, we also investigated the effects of the $\sigma_{s/a}$ values in a range of 0.054 to
0.062 N m$^{-1}$ as discussed in section 3.2. The shadow area in Fig. 12b represents the variation of linear
fit between the measured and predicted $N_{CCN}$. An under- and over-estimated value of 16% ( slope=0.84)
and 8% (slope=1.08) was obtained for the predicted $N_{CCN}$ to the measured $N_{CCN}$ using a $\sigma_{s/a}$ value of
0.054 and 0.062 N m$^{-1}$ respectively, indicating that the predicted $N_{CCN}$ agreed reasonably with the
measured ones when the $\sigma_{s/a}$ values between 0.054 and 0.062 N m$^{-1}$ were used in this study. We
conclude that the predicted $N_{CCN}$ can agree better with the measured one when including both $\sigma_{s/a}^*$ and



$\kappa_{org}{}^{*}$ in the calculation at low SS.
**4 Summary and Conclusions**
The CCN activity is an important parameter that determines the extent to which atmospheric particles
can influence cloud formation. It is hence essential to predict CCN activity so that a quantitative
assessment of atmospheric particles on cloud formation can be made. While numerous studies were
performed to investigate the CCN activity under different atmospheric conditions around the world,
only a few of them were made in the PRD region in China. In this study, several advanced instruments
(i.e., the SMCA, AMS and H-TDMA) were used to respectively measure CCN activity, chemical
composition, and hygroscopicity in PRD during wintertime 2014. Various schemes were proposed to
determine the CCN activity based on the measurements. Here two important properties were
considered when evaluating the CCN activity: the hygroscopic parameter $\kappa$ and the surface tension of
the particles. Three methods (i.e., the SMCA, the AMS+ZSR, and the H-TDMA) were employed to
calculate the $\kappa$ values based on our measurements. The results show that the deviation between $\kappa_{AMS}$
and $\kappa_{CCN}$ became larger at low supersaturation ratios, indicating that aging process of organic
component for larger size particles led to higher hygroscopicity for those particles. The activation curve
became smoother at low SS, which could be partly attributed to the higher heterogeneity of chemical
composition. In general, the Gf-PDF measured by H-TDMA exhibited a bimodal distribution with a
less-hygroscopic mode and a more-hygroscopic mode. The less-hygroscopic mode was more
significant at smaller diameters, indicating a more external mixing for smaller particles, while the
more-hygroscopic increased with diameter and became broader, implying higher hygroscopicity and





more complex chemical composition for larger particles. The shape of activation curve was related to
the σ values of the Gf-PDF. The higher σ values suggest the higher heterogeneity of chemical
composition and smooth activation curve. A κ value of 0.22-0.30 measured by H-TDMA was obtained
for 40-200 nm particles in this study during the measurement period, larger than those previously
measured in the PRD region, which might indicate an increasing mass fraction of nitrate in recent
years.
Organic compounds could influence CCN activity through modifying the hygroscopicity and surface
tension of the particles. The impacts of organics on CCN activity were also investigated in this study.
The increase of organic mass fraction in the particles could lead to the decrease of the aerosol
hygroscopicity and hence increase the $D_{50}$, especially at low supersaturation. In addition, organics
could decrease the surface tension $\sigma_{s/a}$. This could lead to the underestimated CCN activity if pure
water solution is assumed when inverting the H-TDMA data. We evaluated the impact of the surface
tension on the activation ratios over a wide range of $\sigma_{s/a}$ values (0.03-0.07 N m$^{-1}$) for several measured
size particles (40, 80, 110, and 150 nm) and found that a $\sigma_{s/a}$ value of 0.058 N m$^{-1}$ was the best fit
between predicted AR and measured AR, which could then be used to predict the CCN activity in the
PRD region. Based on the hygroscopicity and chemical composition measured in this study, we
proposed several scheme to predict the CCN activity. Overall, the predicted $N_{CCN}$ agreed well with the
measure one. The slope and $R^2$ of $N_{CCN}$ predicted from average data was similar to the $N_{CCN}$ predicted
from individual data. The $N_{CCN}$ obtained from H-TDMA measurement was under-predicted, if pure
water assumption was used and better agreement with the measured values can be achieved by using



the adjusted $\sigma_{s/a}$ (i.e., 0.058 N m$^{-1}$). Similarly, the $N_{CCN}$ predicted from AMS measurement was
underestimated at low supersaturations and overestimated at high supersaturations, due to an
assumption of fixed 0.1 for $\kappa_{org}$ and the external mixing state. Better predicted CCN concentrations can
be obtained by using $\sigma_{s/a}^{*}$ and $\kappa_{org}^{*}$ in the calculation, especially at low supersaturation. For high
supersaturation, the effect of internal mixing assumption should be taken into consideration. We
concluded that better CCN concentrations with the measurements could be achieved by taking the
effects of organic into account on the hygroscopicty, surface tension, and the mixing state of the
particles. More work on the roles of organics on the CCN activity is obviously needed in order to better
understand the impacts of atmospheric particles on cloud formation and hence climate.
**Acknowledgement**
The authors acknowledge the support from the following funding agencies: National Key R&D
Program of China (2016YFC0201901,2017YFC0209500,2016YFC2003305), National Natural
Science Foundation of China (NSFC) (91644225, 21577177, 41775117), and Guangdong provincial
scientific planning project (2014A020216008, 2016B050502005). Support from the Science and
Technology Innovative Research Team Plan of Guangdong Meteorological Bureau (Grant No.201704)
is also acknowledged.

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

| Species | $\kappa$ |
| --- | --- |
| $NH_4NO_3$ | 0.58 |
| $NH_4HSO_4$ | 0.56 |
| $H_2SO_4$ | 0.90 |
| $(NH_4)_2SO_4$ | 0.48 |
| Organics | 0.10 |





1    **Table 2.** Summary of the measured CCN concentration, activation ratio, and $D_{50}$ at the four
2    supersaturations during the campaign.

| SS | | 0.1% | 0.2% | 0.4% | 0.7% |
|---|---|---|---|---|---|
| | Max | 15165 | 19989 | 25964 | 26208 |
| $N_{CCN}$(#/cm$^3$) | Min | 258 | 361 | 408 | 502 |
| | Mean±STD | 3103±1913 | 5095±2972 | 6524±3783 | 7913±4234 |
| | Max | 0.68 | 0.75 | 0.89 | 0.94 |
| Activation | Min | 0.06 | 0.10 | 0.19 | 0.28 |
| Ratio | Mean±STD | 0.26±0.10 | 0.41±0.14 | 0.53±0.15 | 0.64±0.13 |
| | Max | 268.90 | 194.04 | 145.28 | 97.17 |
| $D_{50}$ (nm) | Min | 112.47 | 76.60 | 43.50 | 24.21 |
| | Mean±STD | 156.02±19.48 | 106.66±16.99 | 77.96±14.86 | 58.45±10.68 |





1     **Table 3.** The average σ values of Gf-PDF measured by H-TDMA for the five diameters at four

2     supersaturations. At each SS, the σ values are respectively calculated for the steep and smooth

3     activation.

| SS(%) | 0.1 | | 0.2 | | 0.4 | | 0.7 | |
|---|---|---|---|---|---|---|---|---|
| Dp(nm) | Steep | Smooth | Steep | Smooth | Steep | Smooth | Steep | Smooth |
| 40 | 0.13 | 0.17 | 0.12 | 0.17 | 0.11 | 0.19 | 0.11 | 0.19 |
| 80 | 0.16 | 0.20 | 0.14 | 0.20 | 0.14 | 0.21 | 0.14 | 0.20 |
| 110 | 0.17 | 0.21 | 0.15 | 0.21 | 0.15 | 0.21 | 0.16 | 0.20 |
| 150 | 0.19 | 0.22 | 0.17 | 0.23 | 0.17 | 0.22 | 0.18 | 0.21 |
| 200 | 0.20 | 0.23 | 0.19 | 0.24 | 0.19 | 0.24 | 0.19 | 0.23 |





1      **Table 4.** The schemes used in the $N_{CCN}$ prediction based on H-TDMA measurement.

| Scheme | Method | Slope | $R^2$ |
|---|---|---|---|
| 1 | Real time activation curve | 0.8275 | 0.93 |
| 2 | Average activation curve | 0.8183 | 0.93 |
| 3 | Real time $D_{50}$ | 0.8869 | 0.93 |
| 4 | Average $D_{50}$ | 0.8738 | 0.93 |
| 5 | Real time activation curve using $\sigma_{s/a}^{*}$ | 0.9377 | 0.93 |





1    **Table 5.** The methods used in the $N_{CCN}$ prediction based on AMS measurement.

| Scheme | Method | Slope | $R^2$ |
|---|---|---|---|
| 6 | Real time bulk composition | 0.9859 | 0.91 |
| 7 | Average bulk composition | 1.0108 | 0.91 |
| 8 | Real time size-resolved composition | 0.9721 | 0.87 |
| 9 | Average size-resolved composition | 0.9742 | 0.86 |



FIGURE CAPTIONS
Fig. 1. A schematic representation of $N_{CCN}$ prediction based on H-TDMA and AMS measurements.
Fig. 2. The mass fraction of the bulk NR-PM$_1$ composition (a) and the mass fraction of the
size-resolved composition (b).
Fig. 3. The median and interquartile PNSD and the corresponding κ values obtained from H-TDMA,
AMS, and CCN measurement during the campaign. The κ was pointed against their corresponding
median $D_{50}$ (SMCA and AMS) or measured diameter (H-TDMA). Dot points represent the median
value and the bars represent the interquartile range. The blue, red, and green represent $\kappa_{CCN}$, $\kappa_{AMS,}$ and
$\kappa_{H-TDMA}$ respectively.
Fig. 4. The sized resolved activation ratios measured by SMCA at the four different supersaturations.
Fig. 5. The Gf-PDF as a function of Gf measured by H-TDMA for the five particle diameters (40, 80,
110, 150, 200 nm)
Fig. 6. The relationship between size-resolved mass fraction of organics and $D_{50}$ at the three
supersaturations. The red, blue, and green dots and line represent 0.1%, 0.2%, and 0.4% SS
respectively.
Fig. 7. The predicted activation ratio based on H-TDMA measurement vs. the measured activation ratio
at 0.1%, 0.2%, 0.4% and 0.7% SS for 40, 80, 110, 150 and 200 nm particles. The pure water for surface
tension (0.072 N m$^{-1}$) was assumed when calculating the AR.
Fig. 8. The relative deviation between predicted AR and measured AR at different assumed $\sigma_{s/a}$. The
color code represents $R^2$ between calculated AR and measured AR.
Fig. 9. The predicted activation ratio using new surface tension assumption ($\sigma_{s/a}^{*}$) based on H-TDMA
measurement vs. the measured activation ratio at 0.1%, 0.2%, 0.4% and 0.7% SS for 40, 80, 110, 150



and 200 nm particles.
Fig. 10. The relationship between measured $N_{CCN}$ and predicted $N_{CCN}$ based on scheme 1, 2, 3, 4 and 5.
Fig. 11. The relationship between measured $N_{CCN}$ and predicted $N_{CCN}$ based on scheme 6, 7, 8 and 9.
Fig. 12. The relationship between measured $N_{CCN}$ and predicted $N_{CCN}$ at SS 0.1% based on
size-resolved chemical composition using $\kappa^{*}_{org}$ (a), and $\kappa^{*}_{org}$ and $\sigma^{*}_{s/a}$ (b). The shadow area represents
the variation of the linear fit using the $\sigma_{s/a}$ values between 0.054 and 0.062 N m$^{-1}$.





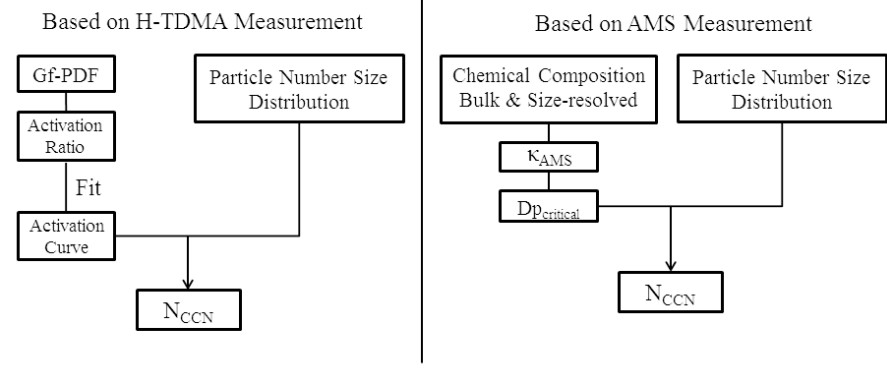

2
3    Fig. 1.



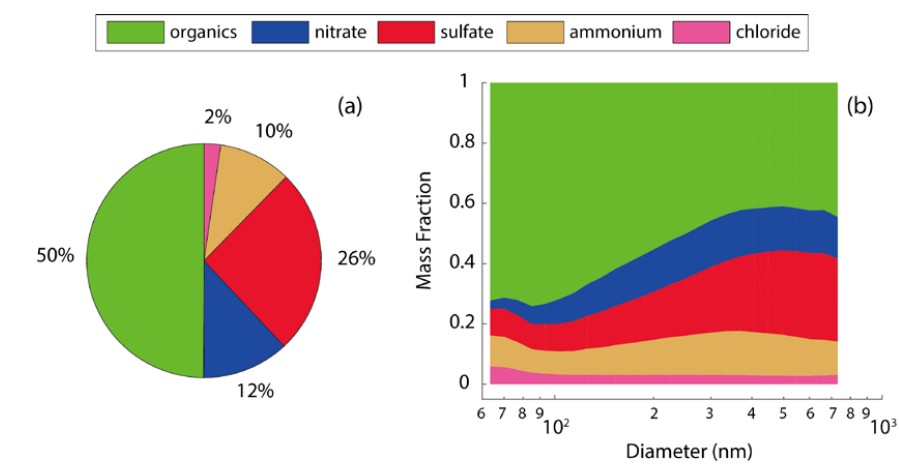

4    Fig. 2.



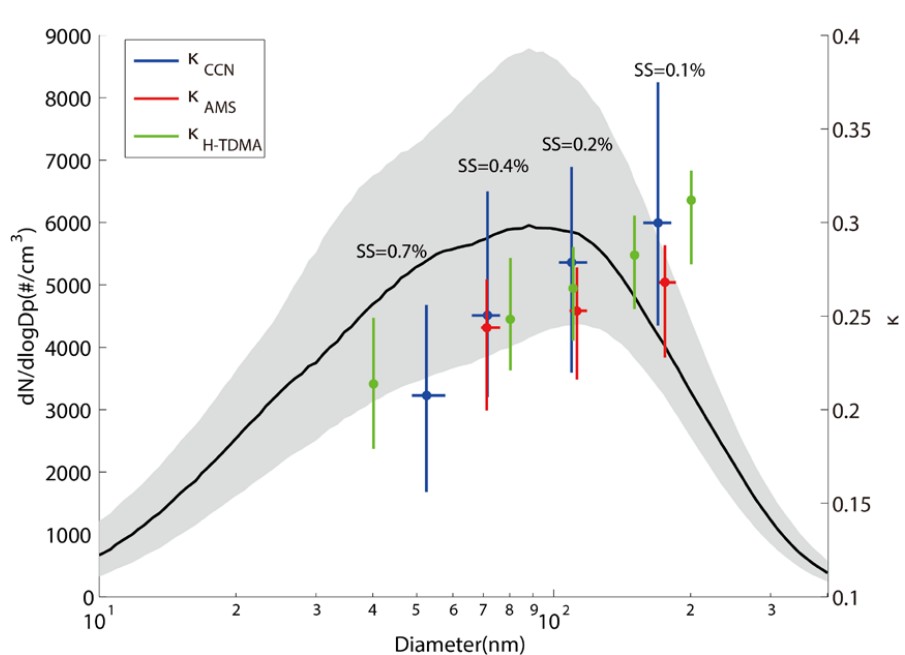

3    Fig. 3.



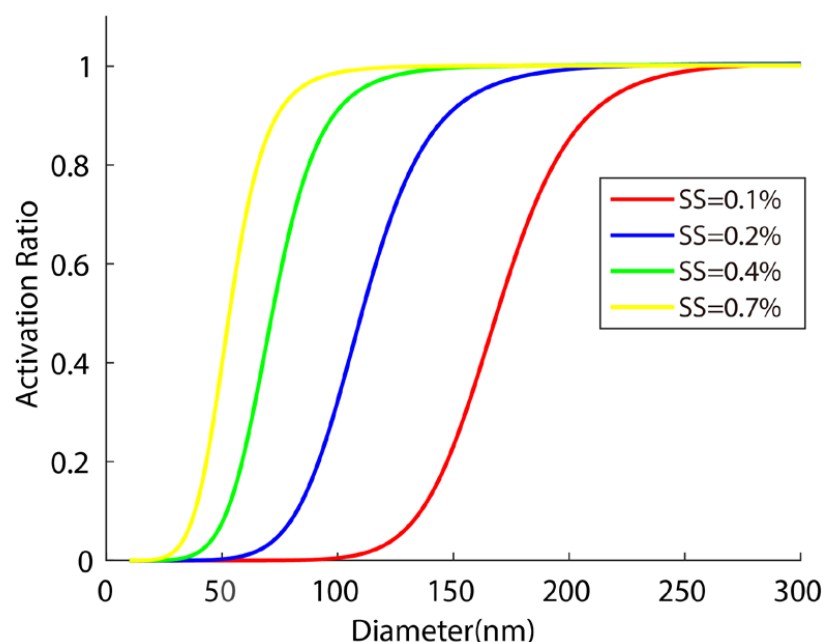

3    Fig. 4.




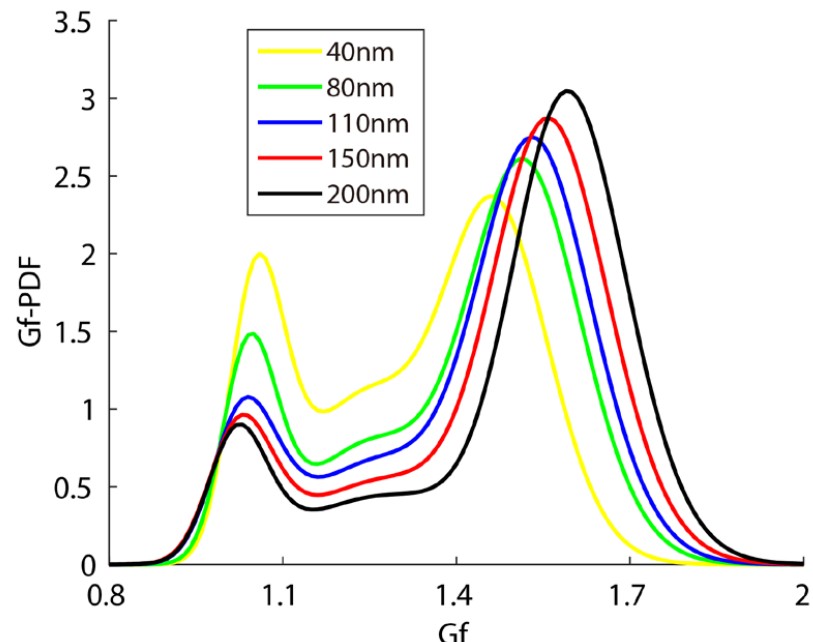

2
3    Fig. 5.
4



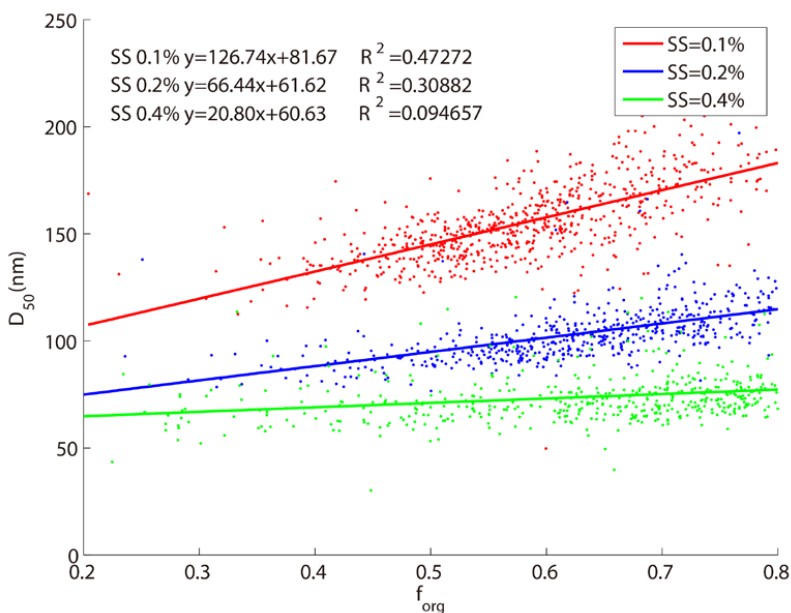

2    Fig. 6.



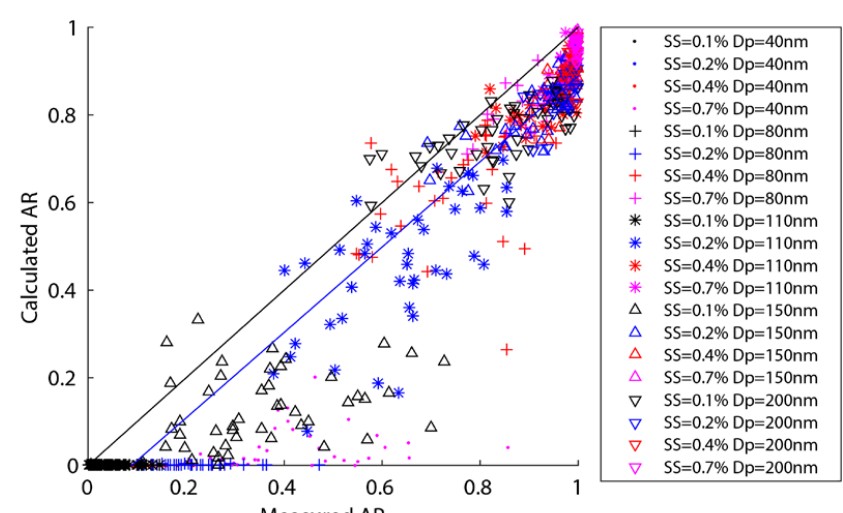

3    Fig. 7.



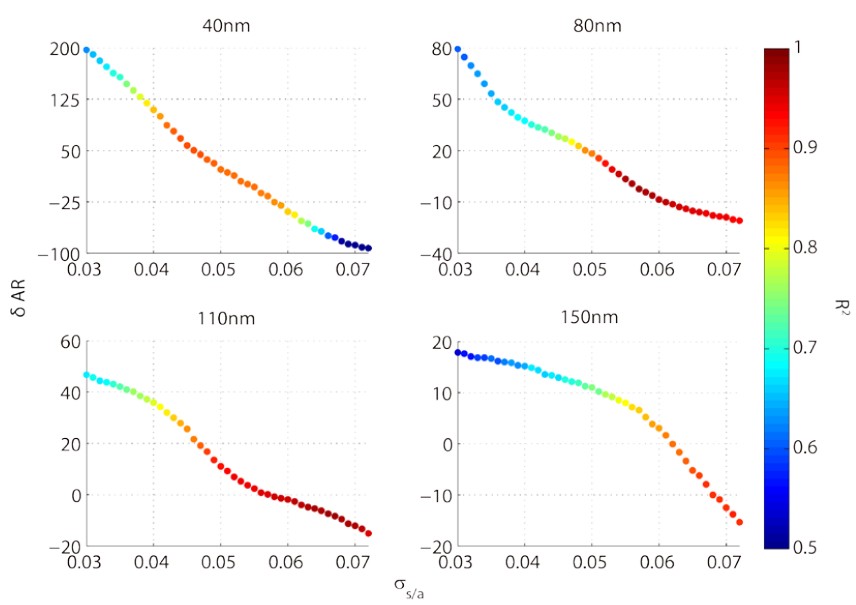

3    Fig. 8.





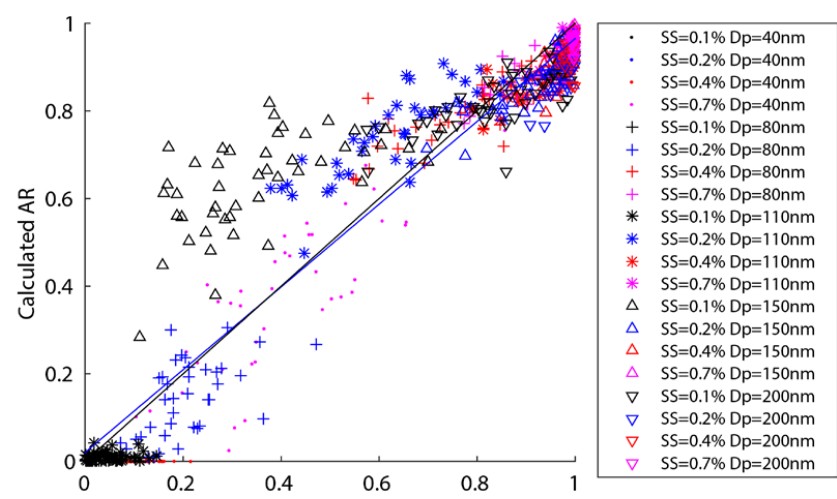

3    Fig. 9.





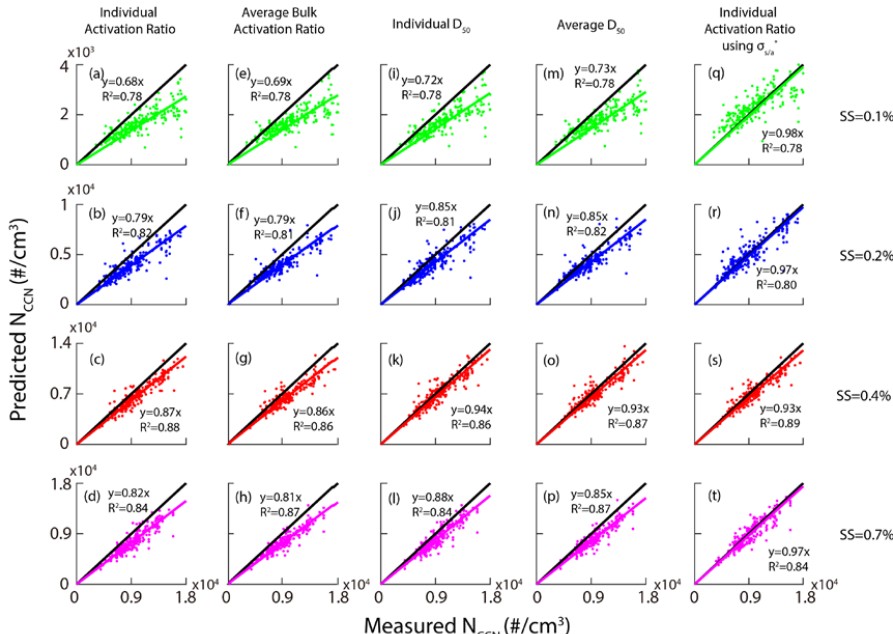

3    Fig. 10.



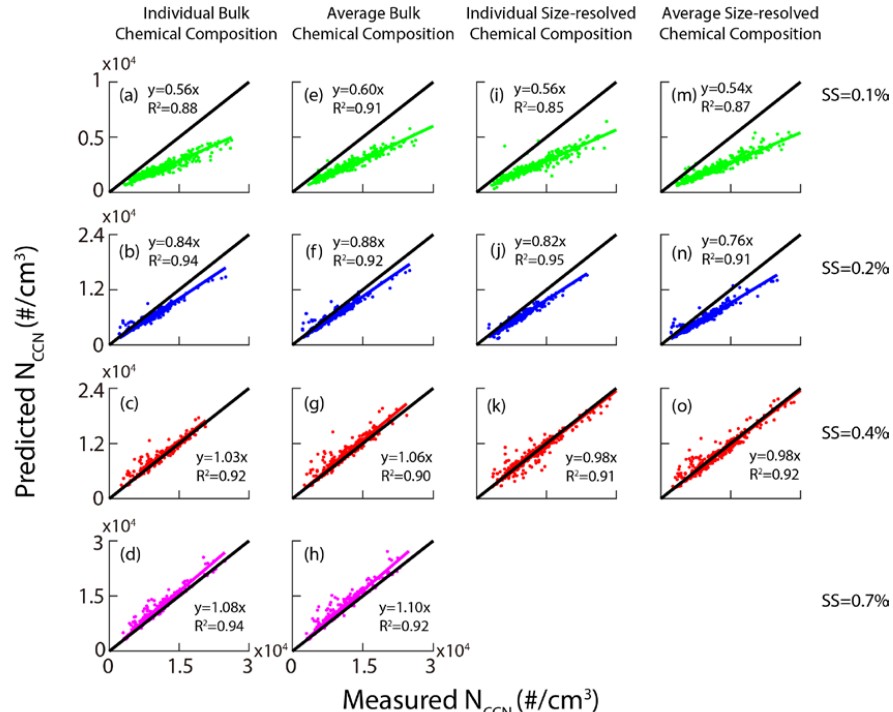

2    Fig. 11.





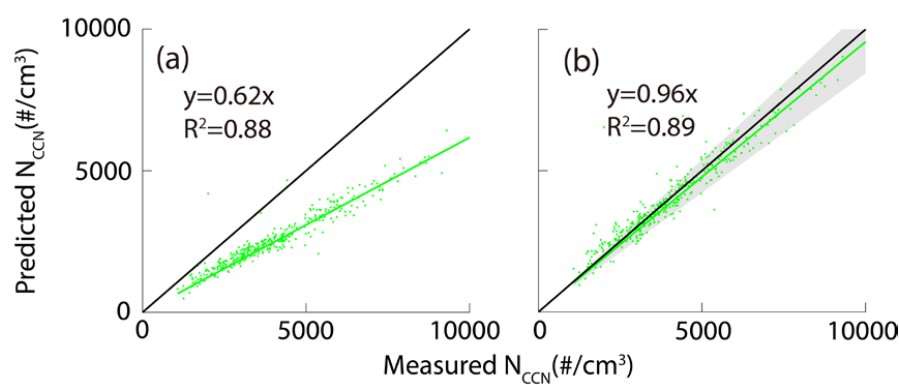

3    Fig. 12.