# Peer review of "The size resolved cloud condensation nuclei (CCN)"

_Atmospheric Chemistry and Physics, 2018_

## Referee Comment (RC1) · Anonymous Referee #3 · 2 Jul 2018

General comments

The manuscript "The size resolved cloud condensation nuclei (CCN) activity and its prediction based on aerosol hygroscopicity and composition in the Pearl Delta River (PRD) Region during wintertime 2014" reports the field observation of CCN activation, hygroscopic growth, and chemical composition of aerosol at a site strongly influenced by anthropogenic emissions in winter 2014. The CCN activation, hygroscopic growth and aerosol chemical composition were measured using scanning mobility CCN analyzer (SMCA), HTDMA and aerosol mass spectrometer (AMS), respectively and $\kappa CCN$, $\kappa HTDMA$, and $\kappa AMS$ were obtained. $\kappa CCN$, $\kappa HTDMA$, and $\kappa AMS$ were compared. The calculated activation ratio (AR) derived from HTDMA was compared with measured AR using CCN and the difference between the calculated AR and measured AR was attributed to the role surface tension change by organics. Moreover, the CCN number concentration ($N_{CCN}$) were derived from HTDMA and AMS using various scheme and compared with measured $N_{CCN}$.

Overall, this study is interesting and provides evaluable data on the CCN activation, hygroscopic growth and aerosol composition as well as links between them. The manuscript fits well the scope of ACP. However, I have some comments before the manuscript is published on ACP. These comments mainly aim to clarify and improve the discussion.

General comments:

1. Regarding methodology, some of the (important) details of are not readily available, which also caused some difficult time (at least for me) understanding some parts of the manuscript.
   For example, the details of how $N_{CCN}$ was derived for each scheme are not available (Pg 22 and Table 4). (Pg 27 lines 19-20) It is not clear what "average data" and "individual data" exactly mean. Were the average data derived from the average over the whole campaign? Does "individual data" mean the same as "real time data"?
   Pg 10 line 12, what reference data of ammonium sulfate were used in the CCN calibration are not provided.
   Pg 12 line 21, have the authors considered or corrected the contribution of double-charged particles? Also in figure 4, are these curve raw data or fitted curve? Have they been corrected for the contribution of double-charged particles? The double-charged particles may contribute to the measured AR in figure 7 and GF-PDF in figure 5.

2. The authors mentioned in the motivation part that "only a few studies were conducted to measure $\kappa$ in the Pearl River Delta" and the site is "an ideal location to investigate the influence of local anthropogenic emissions on the particles properties". However, based on the findings of this study it is not established for me how the results on CCN activation is related to the "speciality"(strong anthropogenic emission) of PRD compared to the observations in other places. It might be worthy discussing this aspect.

3. The difference between the calculated AR and measured AR was attributed to the surface tension change of droplets by organics. I am not sure whether this is robust. There are contributions of other factors. For example, "sparingly soluble" compounds may play a role, which the authors also mentioned. (Pg 20 L18-19) After adjusting the surface tension, the correlation between the calculated AR and measured AR is still not good and quite some data over-estimated the AR. This is also an indication that other factors than the surface tension may play a role.

Specific comments

1. Pg 2 line 2, it was mentioned that "…$\kappa$H-TDMA value was slightly smaller than the $\kappa$CCN one at all diameters…". However, Pg 17, lines 3-4, it was written that "Figure 3 showed that the $\kappa$H-TDMA values were lower than those of the corresponding $\kappa$CCN at most of the SS…". Please clarify the difference of these two statements.

2. Pg 2 line 6, "…be lower than that from the H-TDMA measurement", by "HTDMA measurement" do you mean the CCN measurement? It is somewhat confusing.

3. Pg 2 lines 13-14, "The $N_{CCN}$ values predicted from bulk PM1 were higher (~11.5%)…" It might be better to write "bulk chemical composition of PM1".

4. Pg 6 lines 8-10, "… CCN activity that was characterized by three important parameters: activation diameter (D50), CCN number concentration ($N_{CCN}$), and activation ratio (AR). " The CCN activity actually depends on the chemical composition and particle size(Farmer, Cappa et al. 2015), but is not characterized by CCN number concentrations.

5. Pg 8 lines 1-2, the understanding of relationship between the CCN activity and its controlling factors seems to be not much related to the policy-making on air pollution control. Maybe it is related to the climate-related policy-making.

6. Pg 9 line 10, was neutralizer also used before the 2$^{nd}$ DMA?

7. Pg 16 L8-10, "Aerosol particles with larger sizes were more readily exposed to complex atmospheric composition during their aging process...",  it is not clear why "larger particles were more exposed to complex atmospheric composition during their aging process". Also see Pg 26 L15-16.

8. Pg 17 L7-8, is the difference between $\kappa_{AMS}$ and $\kappa_{CCN}$ and $\kappa_{H-TDMA}$ statistically significant? It might be helpful to comment this.

9. Pg 17 L15-17,  has the fraction of less hygroscopic compounds (eg. organics) decreased together with the increase of nitrate fraction?

10. Pg 18 L17-19, it is not clear for me why the decrease of GF of less-hygroscopic mode and increase of more-hygroscopic mode with increasing particle size indicate that large particles are "tend to be internally mixed". Don't they indicate the larger particles are more externally mixed?

11. Fig. 5, why does the GF of more-hygroscopic mode decrease with increasing particle size and increase for less-hygroscopic mode?

12. Pg 19 L6, it would be helpful to elaborate the parameter C in the method part.

13. Pg 21 L18-19, as mentioned in the general comments, the correlation between the calculated AR and measured AR is still not good after adjusting the surface tension of cloud droplets.

14. Pg 26 L16-18, "…which could be partly attributed to the higher heterogeneity of chemical composition", I guess this refer to the chemical composition of *larger particles*.

15. Table 1, the kappa value of ammonium sulfate is 0.48, which is different from the values compiled in Petters and Kreidenweis (2007). How much would this difference contribute to the different between kappa(AMS) and kappa(CCN) as well as between $N_{CCN}$ and $N_{AMS}$?

16. Pg 25 L8-9, the authors "further assumed the $\kappa_{org}$ values to be 0.15 and 0.1 respectively for particles larger and smaller than 100 nm". Based on the hygroscopic growth measurement, the less hygroscopic mode is likely attributed to be organics. If so, the $\kappa_{org}$ of larger particles should be lower than the smaller particles.  Several studies reported the $\kappa_{org}$ of larger particles are lower than that for smaller particles (Lance, Raatikainen et al. 2013; Zhao, Buchholz et al. 2015). Since hygroscopicity of organics is often found to be related to its chemical composition ($f_{44}$ or O/C) in both field and laboratory studies (Chang, Slowik et al. 2010; Massoli, Lambe et al. 2010; Lambe, Onasch et al. 2011; Mei, Setyan et al. 2013), and others reference therein),  more analysis of AMS data may help validate this assumption of the dependence of $\kappa_{org}$ on particle size.

Technical comments

1. Sect. 2.3.4 should be numbered as 2.3.3.
2. Pg 15 L1, by Eq. 4 do you mean Eq. 5?
3. Pg 15 L5, "…from 69% at…", it looks like the value is more than 70%.
4. Pg 25 L14-15, what does the value 44% refer to?

**References**

Chang, R. Y. W., J. G. Slowik, et al. (2010). "The hygroscopicity parameter (kappa) of ambient organic aerosol at a field site subject to biogenic and anthropogenic influences: relationship to degree of aerosol oxidation." Atmospheric Chemistry and Physics **10**(11): 5047-5064.

Farmer, D. K., C. D. Cappa, et al. (2015). "Atmospheric Processes and Their Controlling Influence on Cloud Condensation Nuclei Activity." Chemical Reviews **115**(10): 4199-4217.

Lambe, A. T., T. B. Onasch, et al. (2011). "Laboratory studies of the chemical composition and cloud condensation nuclei (CCN) activity of secondary organic aerosol (SOA) and oxidized primary organic aerosol (OPOA)." Atmospheric Chemistry and Physics **11**(17): 8913-8928.

Lance, S., T. Raatikainen, et al. (2013). "Aerosol mixing state, hygroscopic growth and cloud activation efficiency during MIRAGE 2006." Atmospheric Chemistry and Physics **13**(9): 5049-5062.

Massoli, P., A. T. Lambe, et al. (2010). "Relationship between aerosol oxidation level and hygroscopic properties of laboratory generated secondary organic aerosol (SOA) particles." Geophysical Research Letters **37**.

Mei, F., A. Setyan, et al. (2013). "CCN activity of organic aerosols observed downwind of urban emissions during CARES." Atmospheric Chemistry and Physics **13**(24): 12155-12169.

Petters, M. D. and S. M. Kreidenweis (2007). "A single parameter representation of hygroscopic growth and cloud condensation nucleus activity." Atmospheric Chemistry and Physics **7**(8): 1961-1971.

Zhao, D. F., A. Buchholz, et al. (2015). "Size-dependent hygroscopicity parameter ($\kappa$) and chemical composition of secondary organic cloud condensation nuclei." Geophysical Research Letters **42**(24): 10920-10928.

---

## Referee Comment (RC2) · Anonymous Referee #1 · 26 Aug 2018

**Summary:**

This work demonstrates the field measurement results of CCN activity, hygroscopicity and chemical compositions of aerosol particles in the PRD region. The manuscript fits well to the scope of ACP and presents valuable results. Thus I recommend it to be published after the following comments listed below have been adequately addressed.

**Comments:**

1. Section 2.2.2: Please give more information of reference data used in the köhler theory when performing the CCNC calibration. This is very important because different parameterizations will retrieve different critical supersaturations (Rose et al., 2008;Wang et al., 2017).

2. Section 2.3.4: It should be section 2.3.3. Still, I am confused with the method to perform the CCN prediction based on HTDMA data. I would suggest the authors give an exemplary case either in the text or supplement referring to Lukas et al., (2010).

3. Table 1: where these data come from, please add reference. I guess these kappa data are retrieved with T of 298.15 K. But in Section 2.3.1, the T you used is 293 K, why?

4. Table 2: Based on the SMCA measurements, you should get size-resolved activation ratio, so I do not understand the max. and min. values of AR here? In principle, it should be 1 and 0. I guess you calculate the overall AR, please clarify and explain the reason why you put it here.

5. Page 16, line 8-10: Please explain this sentence, it is not clear.

6. Page 17, line3-4: This is not consistence with the statement in the abstract, please revise.

7. Page 17, line 5-7: The difference between kappa-CCN and kappa-HTDMA may also due to the parameterizations used in the CCNC and HTDMA calibration. See Wang et al., (2017). Please consider it and give more information as suggested

in comment 1.

8. Page 17, line 11-12: Any evidence? I guess the larger hygroscopicity is mainly due to the bigger particle size.

9. Figure 4, have you corrected the double charge effect of DMA?

10. Page 17, line 16-19: This sentence ("the peak in the less-hygroscopic mode declined … while the one in the more-hygroscopic mode climbed…") is not clear. Do you mean the relative fraction of less-hygroscopic compounds decreased and more-hygroscopic compounds increased at larger particle size?

11. Figure 7: I am not quite sure that the impacts of organics can fully explain the difference between the calculated and measured AR. The bias is still obvious even the configured surface tension (0.072) is used, indicating the other factors should also be considered. Many studies (Petters et al., 2009;Wex et al., 2009;Hersey et al., 2013;Wu et al., 2013;Hong et al., 2014;Hansen et al., 2015;Mikhailov et al., 2015;Pajunoja et al., 2015;Zhao et al., 2016) have reported the different hygroscopic properties from CCNC and HTDMA measurements. I would suggest more discussions should be added here.

12. Page 20, line 20-21: Add reference.

13. Figure 9: please provide $R^2$.

14. Figure 10 and 11: what dose the black line mean? Is it 1:1 line? then the scale should be checked.

15. There are several grammar mistakes in the text, the language and symbols should be checked carefully once more before publication.

References:

Hansen, A. M. K., Hong, J., Raatikainen, T., Kristensen, K., Ylisirniö, A., Virtanen, A., Petäjä, T., Glasius, M., and Prisle, N. L.: Hygroscopic properties and cloud condensation nuclei activation of limonene-derived organosulfates and their mixtures with ammonium sulfate, Atmos Chem Phys, 15, 14071-14089, 10.5194/acp-15-14071-2015, 2015.

Hersey, S. P., Craven, J. S., Metcalf, A. R., Lin, J., Lathem, T., Suski, K. J., Cahill, J. F., Duong, H. T., Sorooshian, A., Jonsson, H. H., Shiraiwa, M., Zuend, A., Nenes, A., Prather, K. A., Flagan, R. C., and Seinfeld, J. H.: Composition and hygroscopicity of the Los Angeles Aerosol: CalNex, J Geophys Res-Atmos, 118, 3016-3036, 10.1002/jgrd.50307, 2013.

Hong, J., Häkkinen, S. A. K., Paramonov, M., Äijälä, M., Hakala, J., Nieminen, T., Mikkilä, J., Prisle, N. L., Kulmala, M., Riipinen, I., Bilde, M., Kerminen, V. M., and Petäjä, T.: Hygroscopicity, CCN and volatility

properties of submicron atmospheric aerosol in a boreal forest environment during the summer of 2010, Atmos Chem Phys, 14, 4733-4748, 10.5194/acp-14-4733-2014, 2014.

Lukas, K., Martin, G., Ernest, W., Hanna, H., J., C. D., Thomas, H., Birgitta, S., Almut, A., and Urs, B.: Subarctic atmospheric aerosol composition: 3. Measured and modeled properties of cloud condensation nuclei, Journal of Geophysical Research: Atmospheres, 115, doi:10.1029/2009JD012447, 2010.

Mikhailov, E. F., Mironov, G. N., Pöhlker, C., Chi, X., Krüger, M. L., Shiraiwa, M., Förster, J. D., Pöschl, U., Vlasenko, S. S., Ryshkevich, T. I., Weigand, M., Kilcoyne, A. L. D., and Andreae, M. O.: Chemical composition, microstructure, and hygroscopic properties of aerosol particles at the Zotino Tall Tower Observatory (ZOTTO), Siberia, during a summer campaign, Atmos Chem Phys, 15, 8847-8869, 10.5194/acp-15-8847-2015, 2015.

Pajunoja, A., Lambe, A. T., Hakala, J., Rastak, N., Cummings, M. J., Brogan, J. F., Hao, L., Paramonov, M., Hong, J., Prisle, N. L., Malila, J., Romakkaniemi, S., Lehtinen, K. E. J., Laaksonen, A., Kulmala, M., Massoli, P., Onasch, T. B., Donahue, N. M., Riipinen, I., Davidovits, P., Worsnop, D. R., Petäjä, T., and Virtanen, A.: Adsorptive uptake of water by semisolid secondary organic aerosols, Geophys. Res. Lett., n/a-n/a, 10.1002/2015GL063142, 2015.

Petters, M. D., Wex, H., Carrico, C. M., Hallbauer, E., Massling, A., McMeeking, G. R., Poulain, L., Wu, Z., Kreidenweis, S. M., and Stratmann, F.: Towards closing the gap between hygroscopic growth and activation for secondary organic aerosol – Part 2: Theoretical approaches, Atmos Chem Phys, 9, 3999-4009, 10.5194/acp-9-3999-2009, 2009.

Rose, D., Gunthe, S. S., Mikhailov, E., Frank, G. P., Dusek, U., Andreae, M. O., and Pöschl, U.: Calibration and measurement uncertainties of a continuous-flow cloud condensation nuclei counter (DMT-CCNC): CCN activation of ammonium sulfate and sodium chloride aerosol particles in theory and experiment, Atmos Chem Phys, 8, 1153-1179, 10.5194/acp-8-1153-2008, 2008.

Wang, Z., Cheng, Y., Ma, N., Mikhailov, E., Pöschl, U., and Su, H.: Dependence of the hygroscopicity parameter κ on particle size, humidity and solute concentration: implications for laboratory experiments, field measurements and model studies, Atmos. Chem. Phys. Discuss., 2017, 1-33, 10.5194/acp-2017-253, 2017.

Wex, H., Petters, M. D., Carrico, C. M., Hallbauer, E., Massling, A., McMeeking, G. R., Poulain, L., Wu, Z., Kreidenweis, S. M., and Stratmann, F.: Towards closing the gap between hygroscopic growth and activation for secondary organic aerosol: Part 1 – Evidence from measurements, Atmos Chem Phys, 9, 3987-3997, 10.5194/acp-9-3987-2009, 2009.

Wu, Z. J., Poulain, L., Henning, S., Dieckmann, K., Birmili, W., Merkel, M., van Pinxteren, D., Spindler, G., Müller, K., Stratmann, F., Herrmann, H., and Wiedensohler, A.: Relating particle hygroscopicity and CCN activity to chemical composition during the HCCT-2010 field campaign, Atmos Chem Phys, 13, 7983-7996, 10.5194/acp-13-7983-2013, 2013.

Zhao, D. F., Buchholz, A., Kortner, B., Schlag, P., Rubach, F., Fuchs, H., Kiendler-Scharr, A., Tillmann, R., Wahner, A., Watne, Å. K., Hallquist, M., Flores, J. M., Rudich, Y., Kristensen, K., Hansen, A. M. K., Glasius, M., Kourtchev, I., Kalberer, M., and Mentel, T. F.: Cloud condensation nuclei activity, droplet growth kinetics, and hygroscopicity of biogenic and anthropogenic secondary organic aerosol (SOA), Atmos Chem Phys, 16, 1105-1121, 10.5194/acp-16-1105-2016, 2016.

---

## Author Comment (AC1) · 30 Sep 2018

We would like to thank the referee for providing valuable comments on our manuscript and we have carefully addressed the referee's comments point-by-point as follows (referee's comments in black and our responses in red):

Referee's general comments:
1. Regarding methodology, some of the (important) details of are not readily available, which also caused some difficult time (at least for me) understanding some parts of the manuscript. For example, the details of how NCCN was derived for each scheme are not available (Pg 22 and Table 4). (Pg 27 lines 19-20) It is not clear what "average data" and "individual data" exactly mean. Were the average data derived from the average over the whole campaign? Does "individual data" mean the same as "real time data"?

**Response:**

The prediction of $N_{CCN}$ using activation curve means that the $N_{CCN}$ was calculated based on eq. 6 (L21 on p.14). The activation curve can either be real time (Scheme 1 in Table 4) or average (Scheme 2 in Table 4). The prediction of $N_{CCN}$ using the $D_{50}$ means that the $N_{CCN}$ was calculated based on eq. 7 (L6 on p.15) from either real time data (Scheme 3 in Table 4) or average data (Scheme 4 in Table 4).The $D_{50}$ was determined from fitting the size-resolved activation ratio by eq. 4 (L14 on p.13). All the above schemes (Schemes 1-4) use an unadjusted surface tension of water ($\sigma_{s/a}$=0.072 N m$^{-1}$). Scheme 5 predicts $N_{CCN}$ using activation curve from real time data and it uses an adjusted surface tension of water ($\sigma_{s/a}^{*}$=0.058 N m$^{-1}$). The 'average data' refer to as the data that average over the whole period of the campaign. The 'individual data' is the same as 'real time data'.

**Changes in the manuscript:**

Now to avoid confusion, we have replaced 'individual data' with 'real time data' in the manuscript. We have now modified the relevant paragraphs in section 2.3.3 (previously misspelled 2.3.4) as follows,

L5-7 on p.14: "Figure 1 is the schematic diagram of the four approaches we followed to predict $N_{CCN}$ based on the above two measured datasets. In the first approach (I in Fig. 1), the mixing state and size dependence were taken into account."

L11-14 on p.14: "A particle with a $\kappa$ value higher than $\kappa_{critical}$(Dp,SS) was considered to be activated as an CCN (Fig. 1a) and the shadow area represented

the particles which can be activated as CCN for a known diameter and SS."

L17 on p.14 to L12 p.15: "This approach is similar to the one employed in Kammermann et al. (2010), however, we used the size-resolved activation ratio ($AR_{SR}$) to calculate the $N_{CCN}$. The $AR_{SR}$ was determined by fitting the $AR(Dp,SS)$ to the diameter Dp using eq. 4 for the five measured diameters (Fig. 1d). Thus, the calculated $N_{CCN}$ using the activation ratio can be expressed as (Fig. 1e):

$$N_{CCN}(SS) = \int_0^{\infty} AR_{SR}(Dp, SS)N_{CN}(Dp)dDp \tag{6}$$

In the second approach (II in Fig.1), the particles were assumed to be internally mixed. The $D_{50}$ was determined by fitting the $AR(Dp,SS)$ to the diameter Dp (Fig. 1d). The $N_{CCN}$ was obtained by integrating the cloud nuclei concentration for particles larger than $D_{50}$ based on the particle size distribution (Fig. 1f), according to the following equation (eq. 7):

$$N_{CCN}(SS) = \int_{D_{50}}^{\infty} N_{CN}(Dp)dDp \tag{7}$$

In the third and fourth approaches (III and IV in Fig.1), the particles were also assumed to be internally mixed. We then calculated the κ value according to the ZRS rule (eq. 8) based on the AMS measurements.

$$\kappa = \sum_i \varepsilon_i \kappa_i \tag{8}$$

where $\varepsilon_i$ is the volume fraction of each component in the particles, $\kappa_i$ is the κ value of each component."

L6-10 on p.16: "Here instead of being determined from fitting of $AR_{SR}$ to Dp used in the second approach, the $D_{50}$ was calculated from the above κ values using eq. 5. In the third approach, the κ values were size-resolved because the chemical composition of the particles was size dependent (Fig. 1b). In the fourth approach, the particles were assumed to have the same chemical composition and hygroscopicity as those in $PM_1$ (Fig. 1c). The $N_{CCN}$ was then predicted using eq. 7 (Figs. 1g and 1h)."

Since we moved eq. 9 forward to eq. 7, we hence modified the numbers of the subsequent equations in order, i.e., original eq. 7 to eq.8, 8 to 9, etc. We have also modified Fig. 1 for better clarification.

[Figure]

Fig. 1. A schematic representation of $N_{CCN}$ prediction based on the H-TDMA and the AMS measurements. The $N_{CCN}$ can be predicted based on the fitted activation ratio (approach I) and the $D_{50}$ (approach II) both obtained from the H-TDMA measurement, the size-resolved composition (approach III) and the bulk $PM_1$ composition (approach IV) both obtained from the AMS measurement. Panel (a) is the representation of calculating the activation ratio for a specific diameter and SS and the shadow area represents the particles which can be activated as CCN; (b) and (c) are the representations of the κ values obtained respectviely from size-resolved chemical composition and bulk chemical composition; (d) is the reprentation of fitting the activation ratio to the particle diameter Dp (red dot); (e), (f), (g), and (h) are the representations of predicting the $N_{CCN}$ using the four approaches respectviely and the shadow area repsents the particles which can be activated as CCN.

Pg 10 line 12, what reference data of ammonium sulfate were used in the CCN calibration are not provided.

**Response:**
The density and molecular weight of ammonium sulfate were assumed to be 1770 kg m$^{-3}$ and 0.132141 kg mol$^{-1}$, respectively.

**Changes in the manuscript:**
We have added three sentences on L15-20 on p.10 regarding the CCN calibration,

"Previous studied showed that different parameterizations in the Köhler theory can retrieve different critical supersaturations (Rose et al., 2008; Wang et al., 2017). When performing the CCNc calibration, we assumed the density and molecular weight of ammonium sulfate to be 1770 kg m$^{-3}$ and 0.132141 kg mol$^{-1}$, respectively. We also set the temperature and the pressure to 298.15 K and 1026 hPa, respectively. A temperature gradient $\Delta T$ of about 3-8 K in the CCNc column was also used in the calibrations."

Pg 12 line 21, have the authors considered or corrected the contribution of double-charged particles? Also in figure 4, are these curve raw data or fitted curve? Have they been corrected for the contribution of double-charged particles? The double-charged particles may contribute to the measured AR in figure 7 and GF-PDF in figure 5.

**Response:**

The contributions of multiply charged particles were taken into account and we did the multiply charged correction for the SMCA, SMPS and H-TDMA data when the data were inverted. Thus the effects of multiply charged particles in Figs. 5 and 7 have been considered. Figure 4 shows the fitted curve of the activation ratio measured by the SMCA, which has also been corrected for the contribution of doubly charged particles.

**Changes in the manuscript:**

We have now incorporated the above discussion to L16-18 on p.12, "Note that we include multiply charged correction for the SMCA, SMPS and H-TDMA data when the data were inverted so that the contributions of the multiply charged particles were accounted for all the measured particle data." We have modified the caption of Fig. 4 for clarification, "Fig. 4. The sized resolved activation ratios measured by the SMCA at four different supersaturations. Note that the curves were fitted according to the SMCA measurements."

2.  The authors mentioned in the motivation part that "only a few studies were conducted to measure K in the Pearl River Delta" and the site is "an ideal location to investigate the influence of local anthropogenic emissions on the particles properties". However, based on the findings of this studyit is not established for me how the results on CCN activation is related to the "speciality"(strong anthropogenic emission) of PRD compared to the observations in other places. It might be worthy discussing this aspect.

**Response:**

We thank the referee for raising this important issue. Guangzhou locates almost at the center of the PRD region in which the air quality is significantly affected by local anthropogenic emissions. The physical and chemical properties of atmospheric particles in Guangzhou could be different from those in remote areas. Recently, Cai et al. (2017) compared the hygroscopicity and chemical composition between Guangzhou and Cape Hedo in Japan, a marine site rarely affected by human activities. The results showed that the less- and non-hygroscopic modes of atmospheric particles in Cape Hedo could hardly be observed, implying that particles tend to be internally mixed at this marine location. Meanwhile, atmospheric particles in Guangzhou have a higher degree of external mixing because Guangzhou was affected by more anthropogenic emissions than in Cape Hedo. As we presented in this paper, the mixing state could affect the activation ratios and the $N_{CCN}$ prediction. Chemical composition could also affect the CCN activity. The results showed that atmospheric particles were dominated by sulfate in Cape Hedo and organics in Guangzhou, which could lead to different CCN activity. In addition, Mochida et al. (2010) measured the size-resolved CCN activity in Cape Hedo. Their results showed that the activation curve at 0.1% SS in Cape Hedo were steeper than that in Guangzhou. The $D_{50}$ in Cape Hedo was lower than that in Guangzhou, indicating that particles were easier to be activated as CCN in Cape Hedo than in Guangzhou.

**Changes in the manuscript:**

We have added several sentences in two paragraphs to show the effect of anthropogenic emissions on the CCN activity,

L7-10 on p.17: "Mochida et al. (2010) measured the size-resolved CCN activity in Cape Hedo, a remote marine site rarely affected by anthropogenic emissions. The results showed that the $D_{50}$ at 0.1% SS in Cape Hedo was about 130 nm, much larger than that in Guangzhou, leading to higher hygroscopicity of atmospheric particles in Cape Hedo than that in Guangzhou."

L20 on p. 21 to L3 on p.22: "Cai et al. (2017) compared the Gf-PDF between Guangzhou and Cape Hedo and the results showed that only more-hygroscopic (MH) particles were observed in Cape Hedo, indicating that atmospheric particles tend to be more internally mixed in Cape Hedo than in Guangzhou. Meanwhile,

atmospheric particles in Guangzhou have a higher degree of external mixing affected by more anthropogenic emissions, which in turn affect the CCN activity."

3. The difference between the calculated AR and measured AR was attributed to the surface tension change of droplets by organics. I am not sure whether this is robust. There are contributions of other factors. For example, "sparingly soluble" compounds may play a role, which the authors also mentioned. (Pg 20 L18-19) After adjusting the surface tension, the correlation between the calculated AR and measured AR is still not good and quite some data over-estimated the AR. This is also an indication that other factors than the surface tension may play a role.

**Response:**

The referee raised a very good point on the factors which affect the agreement between the calculated AR and the measured AR. The H-TDMA measures the hygroscopicity of the particles regardless of their compositions. For particles containing non-hygroscopic compounds, e.g., the sparingly soluble compounds, a unity of growth factor was measured in the H-TDMA, while for those with hygroscopic compounds, a growth factor greater than unity was measured. Thus the impact of the sparingly soluble compounds in the particles was taken into account when the AR was calculated. However, we agree that factors other than surface tension may also play roles. Previous studies found that the hygroscopicity of the particles measured by the H-TDMA could be lower than that measured by the CCNc (Chan et al., 2008; Pajunoja et al. 2015; Petters et al., 2009; Hansen et al., 2015; Hong et al., 2014) which might be attributed to low soluble compounds in the particles. However, the contributions of low soluble compounds cannot currently be quantitatively evaluated. Laboratory experiments showed that the CCN activity would increase by adding organics to sulfate ammonium particles (Engelhart et al., 2008). Our study found that the calculated AR values were systematically lower than the measured ones if a value of 0.072 N m$^{-1}$ was used for the surface tension. We hence believe that the surface tension might play a major role in the AR prediction than other factors, i.e. the effect of "sparingly soluble" compounds. Indeed, as we showed in the paper, the R$^2$ of the fitting between the measured AR and the calculated AR became higher and the $\delta_{AR}$ approached to zero by adjusting the surface tension from 0.072 N m$^{-1}$ to 0.058 N m$^{-1}$ (Fig. 8), suggesting that the AR prediction could be greatly improved by simply adjusting the value of the surface tension. Because we use a single $\sigma_{s/a}$ value (i.e., 0.058 N m$^{-1}$) for all size particles in the AR prediction, it could lead to

over- or under-prediction of the AR for a specific particle size. For example, the AR for 150 nm particles at 0.1% SS was overestimated by using a $\sigma_{s/a}$ value of 0.058 N m$^{-1}$.

**Changes in the manuscript:**

We have added several sentences to discuss the factors that could cause the difference between the calculated AR and measured AR in L7-14 on p.23,

"Note that the surface tension is not the only factor that determines the AR and other factors such as the sparingly soluble compounds in the particles may contribute to the AR, although they are currently not understood. Previous studies found that the hygroscopicity of the particles measured by the H-TDMA could be lower than that measured by the CCNc (Chan et al., 2008; Pajunoja et al. 2015; Petters et al., 2009; Hansen et al., 2015;Hong et al., 2014) which might be attributed to low soluble compounds in the particles. The deviation of the calculated AR from the measured AR is probably dependent on the degree of dissolution of particles and the oxidative state of the organics in the particles."

Specific comments:

1. Pg 2 line 2, it was mentioned that "…$\kappa$H-TDMA value was slightly smaller than the $\kappa$CCN one at all diameters…". However, Pg 17, lines 3-4, it was written that "Figure 3 showed that the $\kappa$HTDMA values were lower than those of the corresponding $\kappa$CCN at most of the SS…". Please clarify the difference of these two statements.

   In Fig. 3, the $D_{50}$ at a specific supersaturation was the fitted parameter from eq. 4 based on the CCN measurements. For clarification, we have changed the sentence " …the corresponding $\kappa_{CCN}$ at most of the SS…" to   " …the corresponding $\kappa_{CCN}$ at most of the particle sizes…"(L12 on p. 18).

2. Pg 2 line 6, "…be lower than that from the H-TDMA measurement", by "HTDMA measurement", do you mean the CCN measurement? It is somewhat confusing.

   The sentence has been revised to "…be lower than that from the CCN measurements" (L6 on p.2)

3. Pg 2 lines 13-14, "The NCCN values predicted from bulk PM1 were higher (~11.5%)…" It might be better to write "bulk chemical composition of PM1".

The sentence has been revised to "The $N_{CCN}$ values predicted from bulk chemical composition of PM1 were higher (~11.5%)…" (L13-14 on p.2)

4. Pg 6 lines 8-10, "… CCN activity that was characterized by three important parameters: activation diameter (D50), CCN number concentration (NCCN), and activation ratio (AR). " The CCN activity actually depends on the chemical composition and particle size (Farmer, Cappa et al. 2015), but is not characterized by CCN number concentrations.

The sentence has been revised to "… CCN activity that was characterized by two important parameters: activation diameter ($D_{50}$) and activation ratio (AR)." (L8-9 on p.6)

5. Pg 8 lines 1-2, the understanding of relationship between the CCN activity and its controlling factors seems to be not much related to the policy-making on air pollution control. Maybe it is related to the climate-related policy-making.

The "air pollution control" has been changed to "climate-related policy-making" (L2 on p.8)

6. Pg 9 line 10, was neutralizer also used before the 2nd DMA?

The particles were charged in the inlet before it enters the first DMA. There was no neutralizer before the second DMA.

7. Pg 16 L8-10, "Aerosol particles with larger sizes were more readily exposed to complex atmospheric composition during their aging process...", it is not clear why "larger particles were more exposed to complex atmospheric composition during their aging process". Also see Pg 26 L15-16.

After careful consideration, we found the first sentence did not contribute substantially to the main point in Fig. 2. We therefore deleted it to avoid confusion. We have modified the second sentence for clarification (L19-21 on p.29), "The results show that the deviation between $\kappa_{AMS}$ and $\kappa_{CCN}$ became larger at low supersaturation ratios, indicating that the organic components in larger size particles were more aged and hygroscopic."

8. Pg 17 L7-8, is the difference between $\kappa_{AMS}$ and $\kappa_{CCN}$ and $\kappa_{H-TDMA}$ statistically significant? It might be helpful to comment this.

We have added a sentence in L9-11 on p. 18, "As shown in Fig. 3, the difference between $\kappa_{CCN}$ and $\kappa_{H\text{-}TDMA}$ is statistically insignificant at all employed diameters, while the one between $\kappa_{AMS}$ and $\kappa_{CCN}$ became statistically significant at larger sizes of the particles."

9. Pg 17 L15-17, has the fraction of less hygroscopic compounds (eg. organics) decreased together with the increase of nitrate fraction?

We have revised the sentence in L10-14 on p.19, "..., suggesting an increase of the aerosol hygroscopicity, which might result from an increasing mass fraction of nitrate in recent years (Zhang et al., 2015; Itahashi et al., 2018), although the fraction decrease of less hygroscopic compounds is not as significant as the fraction increase of the nitrate. However, the fraction of the non-hydroscopic compounds (i.e. EC) decreases more rapidly than the organic compounds."

10. Pg 18 L17-19, it is not clear for me why the decrease of GF of less-hygroscopic mode and increase of more-hygroscopic mode with increasing particle size indicate that large particles are "tend to be internally mixed". Don't they indicate the larger particles are more externally mixed?

We have added several sentences after "…, indicating that larger particles tend to be internally mixed"(L19 on p.20 to L3 on p.21), "Since less-hygroscopic particles were usually associated with externally mixed black carbon (BC) or fresh organics and more-hygroscopic particles usually represent the inorganics matters or BC coated with inorganics matters (internally mixed). The decrease of peak area of less-hygroscopic mode and the increase of more-hygroscopic mode indicate that the number fraction of less-hygroscopic particles decreased while the more-hygroscopic particles fraction increased. Thus, the particles became more internally mixed."

11. Fig. 5, why does the GF of more-hygroscopic mode decrease with increasing particle size and increase for less-hygroscopic mode?

We have added more sentences in L13-19 on p. 20, "Larger size particles contain higher fractions of more-hygroscopic inorganics matters which lead to the increase of Gf of more-hygroscopic mode. The less-hygroscopic mode usually represents externally mixed black carbon or fresh organics. Thus the less-hygroscopic mode for larger size particles more likely represents the

externally mixed non-hygroscopic black carbon with a Gf value of 0.8-1.1, indicating that the Gf of less-hygroscopic mode decreased and that of more-hygroscopic mode increased with the particle diameter (Fig. 5)."

12. Pg 19 L6, it would be helpful to elaborate the parameter C in the method part.

The parameter C has been elaborated in L16-17 on p.13, "A steep activation curve is associated with a small C value."

13. Pg 21 L18-19, as mentioned in the general comments, the correlation between the calculated AR and measured AR is still not good after adjusting the surface tension of cloud droplets..

This issue has been addressed in general comments #3.

14. Pg 26 L16-18, "…which could be partly attributed to the higher heterogeneity of chemical Composition", I guess this refer to the chemical composition of larger particles.

The referee is correct. We have revised this sentence, "The activation curve became smoother at the low SS, which could be partly attributed to the higher heterogeneity of chemical composition for larger particles."(L21 on p.29-L1 on p.30)

15. Table 1, the kappa value of ammonium sulfate is 0.48, which is different from the values compiled in Petters and Kreidenweis (2007). How much would this difference contribute to the different between kappa(AMS) and kappa(CCN) as well as between NCCN and NAMS?

The $\kappa$ value of 0.48 was taken from Topping et al. (2005). Following the referee's suggestion, we add several sentences to discuss the $\kappa_{AMS}$ values and the $N_{CCN}$ calculated using the $\kappa$ value (0.53) of ammonium sulfate from Petters and Kreidenweis (2007) (L17 on p.26 to L1 on p.27), "Note that the impact on the calculated $\kappa_{AMS}$ values and the predicted $N_{CCN}$ was minor using the $\kappa$ value (0.53) of ammonium sulfate from Petters and Kreidenweis (2007). For example, the $\kappa_{AMS}$ values slightly increased from 0.27 to 0.28 at 0.1% SS; the slopes for scheme 6, 8 and 9 in Table 5 slightly increased from 0.9859 to 0.9898, 0.9721 to 0.9834, and 0.9742 to 0.9973, respectively, while the one for scheme 7 did not change."

16. Pg 25 L8-9, the authors "further assumed the $\kappa_{org}$ values to be 0.15 and 0.1 respectively for particles larger and smaller than 100 nm". Based on the hygroscopic growth measurement, the less hygroscopic mode is likely attributed to be organics. If so, the $\kappa_{org}$ of larger particles should be lower than the smaller particles. Several studies reported the $\kappa_{org}$ of larger particles are lower than that for smaller particles (Lance, Raatikainen et al. 2013; Zhao, Buchholz et al. 2015). Since hygroscopicity of organics is often found to be related to its chemical composition (f44 or O/C) in both field and laboratory studies (Chang, Slowik et al. 2010; Massoli, Lambe et al. 2010; Lambe, Onasch et al. 2011; Mei, Setyan et al. 2013), and others reference therein), more analysis of AMS data may help validate this assumption of the dependence of $\kappa_{org}$ on particle size.

**Response:**

We agree with the referee that the less hygroscopic mode is likely attributed to be organics. However, based on our measurements, we believe that the peak of less-hygroscopic mode for larger size particles was smaller, indicating that the number fraction of less-hygroscopic particles was lower. Our results further suggest that the fraction of the hygroscopic organics became higher or the organics were coated with hygroscopic inorganics matters, leading to the increase of hygroscopicity for larger particles. As suggested by the referee, we analyzed the size-resolved f44 of the AMS data. Figure S1 showed that the f44 increased with diameter, indicating that the degree of oxidation of the organics was higher for larger particles. It could also relate to the higher hygroscopicity of organic aerosol for larger particles. Note that the f44 for particle diameters smaller than 100 nm was discarded due to the poor data quality for those particles.

**Changes in the manuscript:**

We have included Fig. S1 in the supplementary material. We have also added several sentences in L20 on p.18 to L6 on p.19, "Previous studies showed that the $\kappa_{org}$ values of larger particles are lower than those for smaller particles (Lance et al., 2013; Zhao et al. 2015) and hygroscopicity of organics is often found to be related to its chemical composition (f44 or O/C) in both field and laboratory studies (Chang et al. 2010; Massoli et al., 2010; Lambe et al., 2011; Mei et al., 2013, and others reference therein). We showed that the f44 increased with the particle size from the AMS data (Fig. S1). Note that the f44 for particle diameters smaller than 100 nm was discarded due to the data quality. The results indicate that the degree of oxidation of the organics was higher for larger size particles and the hygroscopicity for larger particles is higher (Chang et al., 2010)."

[Figure]

Fig. S1. The size-resolved f44 retrieved from AMS data as a function of particle diameter Dp. The error bar for each measured size was shown for f44.

Technical comments:
1. Sect. 2.3.4 should be numbered as 2.3.3.
   It has been revised(L3 on p.14)
2. Pg 15 L1, by Eq. 4 do you mean Eq. 5?
   It has been revised to eq. 5 (L7 on p.16)
3. Pg 15 L5, "…from 69% at…", it looks like the value is more than 70%.
   It has been corrected to 73% (L14 on p.17)
4. Pg 25 L14-15, what does the value 44% refer to?
   L19-20 on p.28, the sentence has been revised, "For example, the underestimate of $N_{CCN}$ decrease from 44% (Fig. 11a) to 4% (Fig. 12b)".

References:

Chan, M. N., Kreidenweis, S. M., and Chan, C. K.: Measurements of the hygroscopic and deliquescence properties of organic compounds of different solubilities in water and their relationship with cloud condensation nuclei activities, Environ. Sci. Technol., 42, 3602, 2008.
Chang, R. Y. W., Slowik, J. G., Shantz, N. C., Vlasenko, A., Liggio, J., Sjostedt, S. J.,

Leaitch, W. R., and Abbatt, J. P. D.: The hygroscopicity parameter ($\kappa$) of ambient organic aerosol at a field site subject to biogenic and anthropogenic influences: relationship to degree of aerosol oxidation, Atmosp. Chem. Phys., 10, 5047-5064, 2010.

Engelhart, G., Asa-Awuku, A., Nenes, A., and Pandis, S.: CCN activity and droplet growth kinetics of fresh and aged monoterpene secondary organic aerosol, Atmos. Chem. Phys., 8, 3937-3949, 2008.

Farmer, D. K., C. D. Cappa, et al. (2015). "Atmospheric Processes and Their Controlling Influence on Cloud Condensation Nuclei Activity." Chemical Reviews **115**(10): 4199-4217.

Hansen, A. M. K., Hong, J., Raatikainen, T., Kristensen, K., Ylisirniö, A., Virtanen, A., Petäjä, T., Glasius, M., and Prisle, N. L.: Hygroscopic properties and cloud condensation nuclei activation of limonene-derived organosulfates and their mixtures with ammonium sulfate, Atmos Chem Phys, 15, 14071-14089, 10.5194/acp-15-14071-2015, 2015.

Hong, J., Häkkinen, S. A. K., Paramonov, M., Äijälä, M., Hakala, J., Nieminen, T., Mikkilä, J., Prisle, N. L., Kulmala, M., and Riipinen, I.: Hygroscopicity, CCN and volatility properties of submicron atmospheric aerosol in a boreal forest environment during the summer of 2010, Atmos. Chem. Phys., 14, 29097-29136, 2014.

Lambe, A. T., T. B. Onasch, et al. (2011). "Laboratory studies of the chemical composition and cloud condensation nuclei (CCN) activity of secondary organic aerosol (SOA) and oxidized primary organic aerosol (OPOA)." Atmospheric Chemistry and Physics **11**(17): 8913-8928.

Lance, S., T. Raatikainen, et al. (2013). "Aerosol mixing state, hygroscopic growth and cloud activation efficiency during MIRAGE 2006." Atmospheric Chemistry and Physics **13**(9): 5049-5062.

Massoli, P., A. T. Lambe, et al. (2010). "Relationship between aerosol oxidation level and hygroscopic properties of laboratory generated secondary organic aerosol (SOA) particles." Geophysical Research Letters **37**.

Mei, F., A. Setyan, et al. (2013). "CCN activity of organic aerosols observed downwind of urban emissions during CARES." Atmospheric Chemistry and Physics **13**(24): 12155-12169.

Mochida, M., Nishita-Hara, C., Kitamori, Y., Aggarwal, S. G., Kawamura, K., Miura, K., and Takami, A.: Size-segregated measurements of cloud condensation nucleus activity and hygroscopic growth for aerosols at Cape Hedo, Japan, in spring 2008, J. Geophys. Res.: Atmos., 115, 2010.

Koehler, K., Kreidenweis, S., DeMott, P., Prenni, A., Carrico, C., Ervens, B., and Feingold, G.: Water activity and activation diameters from hygroscopicity data-Part II: Application to organic species, Atmos. Chem. Phys., 6, 795-809, 2006.

Pajunoja, A., Lambe, A. T., Hakala, J., Rastak, N., Cummings, M. J., Brogan, J. F., Hao, L., Paramonov, M., Hong, J., and Prisle, N. L.: Adsorptive uptake of water by semisolid secondary organic aerosols, Geophys. Res. Lett. 42, 3063-3068,

2015.

Petters, M., and Kreidenweis, S.: A single parameter representation of hygroscopic growth and cloud condensation nucleus activity, Atmos. Chem. Phys., 7, 1961-1971, 2007.

Petters, M. D., Wex, H., Carrico, C. M., Hallbauer, E., Massling, A., McMeeking, G. R., Poulain, L., Wu, Z., Kreidenweis, S. M., and Stratmann, F.: Towards closing the gap between hygroscopic growth and activation for secondary organic aerosol – Part 2: Theoretical approaches, Atmos. Chem. Phys., 9, 3999-4009, 10.5194/acp-9-3999-2009, 2009.

Topping, D. O., McFiggans, G. B., and Coe, H.: A curved multi-component aerosol hygroscopicity model framework: Part 1 – Inorganic compounds, Atmos. Chem. Phys., 5, 1205-1222, 10.5194/acp-5-1205-2005, 2005.

Zhang, X. Y., Wang, J. Z., Wang, Y. Q., Liu, H. L., Sun, J. Y., and Zhang, Y. M.: Changes in chemical components of aerosol particles in different haze regions in China from 2006 to 2013 and contribution of meteorological factors, Atmos. Chem. Phys., 15, 12935-12952, 2015.

Zhao, D. F., A. Buchholz, et al. (2015). "Size-dependent hygroscopicity parameter (κ) and chemicalcomposition of secondary organic cloud condensation nuclei." Geophysical Research Letters **42**(24): 10920-10928.

---

## Author Comment (AC2) · 30 Sep 2018

We would like to thank the referee for providing valuable comments on our manuscript and we have carefully addressed the referee's comments point-by-point as follows (referee's comments in black and our responses in red):

Referee's comments:

1. Section 2.2.2: Please give more information of reference data used in the köhler theory when performing the CCNC calibration. This is very important because different parameterizations will retrieve different critical supersaturations (Rose et al., 2008;Wang et al., 2017).

   **Response:**

   We agree with the referee that different parameterizations will retrieve different critical supersaturations. When performing the CCNc calibration, we include some important reference data for the ammonium sulfate particles that we used in the CCNc calibration. Specifically, the density and molecular weight of ammonium sulfate were assumed to be 1770 kg m$^{-3}$ and 0.132141 kg mol$^{-1}$, respectively.

   **Changes in the manuscript:**

   We have added three sentences on L15-20 on p.10 regarding the CCN calibration, "Previous studied showed that different parameterizations in the Köhler theory can retrieve different critical supersaturations (Rose et al., 2008; Wang et al., 2017). When performing the CCNc calibration, we assumed the density and molecular weight of ammonium sulfate to be 1770 kg m$^{-3}$ and 0.132141 kg mol$^{-1}$, respectively. We also set the temperature and the pressure to 298.15 K and 1026 hPa, respectively. A temperature gradient $\Delta T$ of about 3-8 K in the CCNc column was also used in the calibrations."

2. Section 2.3.4: It should be section 2.3.3. Still, I am confused with the method to perform the CCN prediction based on HTDMA data. I would suggest the authors give an exemplary case either in the text or supplement referring to Lukas et al., (2010).

   **Response:**

   We proposed five schemes to predict the CCN activity as shown in Table 4. The prediction of $N_{CCN}$ using activation curve means that the $N_{CCN}$ was calculated based on eq. 6 (L21 on p.14). The activation curve can either be real time (Scheme 1 in Table 4) or average (Scheme 2 in Table 4). The prediction of $N_{CCN}$ using the $D_{50}$ means that the $N_{CCN}$ was calculated based on eq. 7 (L6 on p.15) from either

real time data (Scheme 3 in Table 4) or average data (Scheme 4 in Table 4).The $D_{50}$ was determined from fitting the size-resolved activation ratio by eq. 4 (L14 on p.13). All the above schemes (Schemes 1-4) use an unadjusted surface tension of water ($\sigma_{s/a}$=0.072 N m$^{-1}$). Scheme 5 predicts $N_{CCN}$ using activation curve from real time data and it uses an adjusted surface tension of water ($\sigma_{s/a}^{*}$=0.058 N m$^{-1}$). The approach we used here is similar to the one employed in Kammermann et al. (2010) as we described the detailed changes below.

**Changes in the manuscript:**

We have now modified the relevant paragraphs in section 2.3.3 (previously misspelled 2.3.4) as follows,

L5-7 on p.14: "Figure 1 is the schematic diagram of the four approaches we followed to predict $N_{CCN}$ based on the above two measured datasets. In the first approach (I in Fig. 1), the mixing state and size dependence were taken into account."

L11-14 on p.14: "A particle with a $\kappa$ value higher than $\kappa_{critical}$(Dp,SS) was considered to be activated as an CCN (Fig. 1a) and the shadow area represented the particles which can be activated as CCN for a known diameter and SS."

L17 on p.14 to L12 p.15: "This approach is similar to the one employed in Kammermann et al. (2010), however, we used the size-resolved activation ratio ($AR_{SR}$) to calculate the $N_{CCN}$. The $AR_{SR}$ was determined by fitting the AR(Dp,SS) to the diameter Dp using eq. 4 for the five measured diameters (Fig. 1d). Thus, the calculated $N_{CCN}$ using the activation ratio can be expressed as (Fig. 1e):

$$N_{CCN}(SS) = \int_0^\infty AR_{SR}(Dp,SS)N_{CN}(Dp)dDp \qquad (6)$$

In the second approach (II in Fig.1), the particles were assumed to be internally mixed. The $D_{50}$ was determined by fitting the AR(Dp,SS) to the diameter Dp (Fig. 1d). The $N_{CCN}$ was obtained by integrating the cloud nuclei concentration for particles larger than $D_{50}$ based on the particle size distribution (Fig. 1f), according to the following equation (eq. 7):

$$N_{CCN}(SS) = \int_{D_{50}}^\infty N_{CN}(Dp)dDp \qquad (7)$$

In the third and fourth approaches (III and IV in Fig.1), the particles were also

assumed to be internally mixed. We then calculated the κ value according to the ZRS rule (eq. 8) based on the AMS measurements.

$$\kappa = \sum_i \varepsilon_i \kappa_i \tag{8}$$

where $\varepsilon_i$ is the volume fraction of each component in the particles, $\kappa_i$ is the κ value of each component."

L6-10 on p.16: "Here instead of being determined from fitting of $AR_{SR}$ to Dp used in the second approach, the $D_{50}$ was calculated from the above κ values using eq. 5. In the third approach, the κ values were size-resolved because the chemical composition of the particles was size dependent (Fig. 1b). In the fourth approach, the particles were assumed to have the same chemical composition and hygroscopicity as those in $PM_1$ (Fig. 1c). The $N_{CCN}$ was then predicted using eq. 7 (Figs. 1g and 1h)."

Since we moved eq. 9 forward to eq. 7, we hence modified the numbers of the subsequent equations in order, i.e., original eq. 7 to eq.8, 8 to 9, etc. We have also modified Fig. 1 for better clarification.

[Figure]

Fig. 1. A schematic representation of $N_{CCN}$ prediction based on the H-TDMA and the AMS measurements. The $N_{CCN}$ can be predicted based on the fitted activation

ratio (approach I) and the $D_{50}$ (approach II) both obtained from the H-TDMA measurement, the size-resolved composition (approach III) and the bulk $PM_1$ composition (approach IV) both obtained from the AMS measurement. Panel (a) is the representation of calculating the activation ratio for a specific diameter and SS and the shadow area represents the particles which can be activated as CCN; (b) and (c) are the representations of the κ values obtained respectviely from size-resolved chemical composition and bulk chemical composition; (d) is the reprentation of fitting the activation ratio to the particle diameter Dp (red dot); (e), (f), (g), and (h) are the representations of predicting the $N_{CCN}$ using the four approaches respectviely and the shadow area repsents the particles which can be activated as CCN.

3. Table 1: where these data come from, please add reference. I guess these kappa data are retrieved with T of 298.15 K. But in Section 2.3.1, the T you used is 293 K, why?

We have included the references in Table 1. We have changed the temperature in Section 2.3.1 to 298.15 K for consistency.

**Table 1.** The κ values of the related species in the study.

| Species | κ |
| --- | --- |
| $NH_4NO_3$ | 0.58[a] |
| $NH_4HSO_4$ | 0.56[a] |
| $H_2SO_4$ | 0.90[a] |
| $(NH_4)_2SO_4$ | 0.48[a] |
| Organics | 0.10[b] |

[a] The κ of inorganics compounds are derived from ADDEM (Topping et al., 2005)

[b] The κ of organics was taken from Meng et al. (2014)

4. Table 2: Based on the SMCA measurements, you should get size-resolved activation ratio, so I do not understand the max. and min. values of AR here? In principle, it should be 1 and 0. I guess you calculate the overall AR, please clarify

and explain the reason why you put it here

The activation ratio (AR) in Table 2 represents the ratio of total number of cloud condensation nuclei ($N_{CCN}$) to the total particle number ($N_{CN, tot}$). The total AR was not only affected by the particle hygroscopicity but also the particle number size distribution (PNSD). For example, if the PNSD was unimodal with a peak at about 100 nm, while the $D_{50}$ was about 180 nm at 0.1% SS, resulting in only a minor fraction of particles that were larger than the $D_{50}$, implying a low value of AR. However, during the pollution events, the PNSD was often boarder and the peak was shifted to a larger size (e.g., 130 nm), leading to a larger AR. To avoid any confusion, we use $N_{CCN}/N_{CN,tot}$ instead of activation ratio to represent the ratio of total $N_{CCN}$ to the total particle number in the manuscript.

5. Page 16, line 8-10: Please explain this sentence, it is not clear.

Since the sentence did not contribute substantially to the main point in Fig. 2, we have removed it from the text to avoid confusion.

6. Page 17, line3-4: This is not consistence with the statement in the abstract, please revise.

In Fig. 3, the $D_{50}$ at a specific supersaturation was the fitted parameter from eq. 4 based on the CCN measurements. For clarification, we have changed the sentence " …the corresponding $\kappa_{CCN}$ at most of the SS…" to " …the corresponding $\kappa_{CCN}$ at most of the particle sizes…"(L12 on p. 18).

7. Page 17, line 5-7: The difference between kappa-CCN and kappa-HTDMA may also due to the parameterizations used in the CCNC and HTDMA calibration. See Wang et al., (2017). Please consider it and give more information as suggested in comment 1.

We agree with the referee. We figured out that the difference between $\kappa_{CCN}$ and $\kappa_{H-TDMA}$ is statistically insignificant. We have added a sentence in L9-11 on p. 18, "As shown in Fig. 3, the difference between $\kappa_{CCN}$ and $\kappa_{H-TDMA}$ is statistically insignificant at all employed diameters, while the one between $\kappa_{AMS}$ and $\kappa_{CCN}$ became statistically significant at larger sizes of the particles."

8. Page 17, line 11-12: Any evidence? I guess the larger hygroscopicity is mainly due to the bigger particle size.

**Response:**

Based on our measurements, we believe that the peak of less-hygroscopic mode for larger size particles was smaller, indicating that the number fraction of less-hygroscopic particles was lower. Our results further suggest that the fraction of the hygroscopic organics became higher or the organics were coated with hygroscopic inorganics matters, leading to the increase of hygroscopicity for larger particles. As suggested by the other referee, we analyzed the size-resolved f44 of the AMS data. Figure S1 showed that the f44 increased with diameter, indicating that the degree of oxidation of the organics was higher for larger particles. It could also relate to the higher hygroscopicity of organic aerosol for larger particles. Note that the f44 for particle diameters smaller than 100 nm was discarded due to the poor data quality for those particles.

**Changes in the manuscript:**

We have included Fig. S1 in the supplementary material. We have also added several sentences in L20 on p.18 to L6 on p.19, "Previous studies showed that the $\kappa_{org}$ values of larger particles are lower than those for smaller particles (Lance et al., 2013; Zhao et al. 2015) and hygroscopicity of organics is often found to be related to its chemical composition (f44 or O/C) in both field and laboratory studies (Chang et al. 2010; Massoli et al., 2010; Lambe et al., 2011; Mei et al., 2013, and others reference therein). We showed that the f44 increased with the particle size from the AMS data (Fig. S1). Note that the f44 for particle diameters smaller than 100 nm was discarded due to the data quality. The results indicate that the degree of oxidation of the organics was higher for larger size particles and the hygroscopicity for larger particles is higher (Chang et al., 2010)."

[Figure]

Fig. S1. The size-resolved f44 retrieved from AMS data as a function of particle diameter Dp. The error bar for each measured size was shown for f44.

9. Figure 4, have you corrected the double charge effect of DMA?

**Response:**

The contributions of multiply charged particles were taken into account and we did the multiply charged correction for the SMCA, SMPS and H-TDMA data when the data were inverted. Thus the effects of multiply charged particles in Figs. 5 and 7 have been considered. Figure 4 shows the fitted curve of the activation ratio measured by the SMCA, which has also been corrected for the contribution of doubly charged particles.

**Changes in the manuscript:**

We have now incorporated the above discussion to L16-18 on p.12, "Note that we include multiply charged correction for the SMCA, SMPS and H-TDMA data when the data were inverted so that the contributions of the multiply charged particles were accounted for all the measured particle data." We have modified the caption of Fig. 4 for clarification, "Fig. 4. The sized resolved activation ratios measured by the SMCA at four different supersaturations. Note that the curves were fitted according to the SMCA measurements."

10. Page 17, line 16-19: This sentence ("the peak in the less-hygroscopic mode declined … while the one in the more-hygroscopic mode climbed…") is not clear. Do you mean the relative fraction of less-hygroscopic compounds decreased and more-hygroscopic compounds increased at larger particle size?

We have added several sentences after "…, indicating that larger particles tend to be internally mixed"(L19 on p.20 to L3 on p.21), "Since less-hygroscopic particles were usually associated with externally mixed black carbon (BC) or fresh organics and more-hygroscopic particles usually represent the inorganics matters or BC coated with inorganics matters (internally mixed). The decrease of peak area of less-hygroscopic mode and the increase of more-hygroscopic mode indicate that the number fraction of less-hygroscopic particles decreased while the more-hygroscopic particles fraction increased. Thus, the particles became more internally mixed."

We believe that we have addressed this issue and we have then deleted the titled sentence for clarification.

11. Figure 7: I am not quite sure that the impacts of organics can fully explain the difference between the calculated and measured AR. The bias is still obvious even the configured surface tension (0.072) is used, indicating the other factors should also be considered. Many studies (Petters et al., 2009;Wex et al., 2009;Hersey et al., 2013;Wu et al., 2013;Hong et al., 2014;Hansen et al., 2015;Mikhailov et al., 2015;Pajunoja et al., 2015;Zhao et al., 2016) have reported the different hygroscopic properties from CCNC and HTDMA measurements. I would suggest more discussions should be added here.

**Response:**

The referee raised a very good point on the factors which affect the agreement between the calculated AR and the measured AR. The H-TDMA measures the hygroscopicity of the particles regardless of their compositions. For particles containing non-hygroscopic compounds, e.g., the sparingly soluble compounds, a unity of growth factor was measured in the H-TDMA, while for those with hygroscopic compounds, a growth factor greater than unity was measured. Thus the impact of the sparingly soluble compounds in the particles was taken into account when the AR was calculated. However, we agree that factors other than surface tension may also play roles. Previous studies found that the hygroscopicity of the particles measured by the H-TDMA could be lower than that measured by the CCNc (Chan et al., 2008; Pajunoja et al. 2015; Petters et al., 2009; Hansen et

al., 2015; Hong et al., 2014) which might be attributed to low soluble compounds in the particles. However, the contributions of low soluble compounds cannot currently be quantitatively evaluated. Laboratory experiments showed that the CCN activity would increase by adding organics to sulfate ammonium particles (Engelhart et al., 2008). Our study found that the calculated AR values were systematically lower than the measured ones if a value of 0.072 N m$^{-1}$ was used for the surface tension. We hence believe that the surface tension might play a major role in the AR prediction than other factors, i.e. the effect of "sparingly soluble" compounds. Indeed, as we showed in the paper, the $R^2$ of the fitting between the measured AR and the calculated AR became higher and the $\delta_{AR}$ approached to zero by adjusting the surface tension from 0.072 N m$^{-1}$ to 0.058 N m$^{-1}$ (Fig. 8), suggesting that the AR prediction could be greatly improved by simply adjusting the value of the surface tension. Because we use a single $\sigma_{s/a}$ value (i.e., 0.058 N m$^{-1}$) for all size particles in the AR prediction, it could lead to over- or under-prediction of the AR for a specific particle size. For example, the AR for 150 nm particles at 0.1% SS was overestimated by using a $\sigma_{s/a}$ value of 0.058 N m$^{-1}$.

**Changes in the manuscript:**
We have added several sentences to discuss the factors that could cause the difference between the calculated AR and measured AR in L7-14 on p.23,
"Note that the surface tension is not the only factor that determines the AR and other factors such as the sparingly soluble compounds in the particles may contribute to the AR, although they are currently not understood. Previous studies found that the hygroscopicity of the particles measured by the H-TDMA could be lower than that measured by the CCNc (Chan et al., 2008; Pajunoja et al. 2015; Petters et al., 2009; Hansen et al., 2015;Hong et al., 2014) which might be attributed to low soluble compounds in the particles. The deviation of the calculated AR from the measured AR is probably dependent on the degree of dissolution of particles and the oxidative state of the organics in the particles."

12. Page 20, line 20-21: Add reference.

   The reference has been added (L17 on p. 23).

13. Figure 9: please provide $R^2$.

The R$^2$ has been added in Figs. 7 and 9.

14. Figure 10 and 11: what dose the black line mean? Is it 1:1 line? then the scale should be checked.

The black lines represent the 1:1 lines. Figure 10 and 11 and their captions have been revised as attached below:

[Figure]

Fig. 10. The relationship between measured $N_{CCN}$ and predicted $N_{CCN}$ based on scheme 1, 2, 3, 4 and 5. The black lines represent 1:1 lines.

[Figure]

Fig. 11. The relationship between measured $N_{CCN}$ and predicted $N_{CCN}$ based on scheme 6, 7, 8 and 9. The black lines represent 1:1 lines.

15. There are several grammar mistakes in the text, the language and symbols should be checked carefully once more before publication.

We have carefully checked the texts to improve the quality of the manuscript.

References

Chan, M. N., Kreidenweis, S. M., and Chan, C. K.: Measurements of the hygroscopic and deliquescence properties of organic compounds of different solubilities in water and their relationship with cloud condensation nuclei activities, Environ. Sci. Technol., 42, 3602, 2008.

Chang, R. Y. W., Slowik, J. G., Shantz, N. C., Vlasenko, A., Liggio, J., Sjostedt, S. J., Leaitch, W. R., and Abbatt, J. P. D.: The hygroscopicity parameter ($\kappa$) of ambient organic aerosol at a field site subject to biogenic and anthropogenic influences: relationship to degree of aerosol oxidation, Atmosp. Chem. Phys., 10, 5047-5064, 2010.

Engelhart, G., Asa-Awuku, A., Nenes, A., and Pandis, S.: CCN activity and droplet growth kinetics of fresh and aged monoterpene secondary organic aerosol, Atmos. Chem. Phys., 8, 3937-3949, 2008.

Farmer, D. K., C. D. Cappa, et al. (2015). "Atmospheric Processes and Their Controlling Influence on Cloud Condensation Nuclei Activity." Chemical Reviews **115**(10): 4199-4217.

Hansen, A. M. K., Hong, J., Raatikainen, T., Kristensen, K., Ylisirniö, A., Virtanen, A., Petäjä, T., Glasius, M., and Prisle, N. L.: Hygroscopic properties and cloud condensation nuclei activation of limonene-derived organosulfates and their mixtures with ammonium sulfate, Atmos Chem Phys, 15, 14071-14089, 10.5194/acp-15-14071-2015, 2015.

Hong, J., Häkkinen, S. A. K., Paramonov, M., Äijälä, M., Hakala, J., Nieminen, T., Mikkilä, J., Prisle, N. L., Kulmala, M., and Riipinen, I.: Hygroscopicity, CCN and volatility properties of submicron atmospheric aerosol in a boreal forest environment during the summer of 2010, Atmos. Chem. Phys., 14, 29097-29136, 2014.

Lambe, A. T., T. B. Onasch, et al. (2011). "Laboratory studies of the chemical composition and cloud condensation nuclei (CCN) activity of secondary organic aerosol (SOA) and oxidized primary organic aerosol (OPOA)." Atmospheric Chemistry and Physics **11**(17): 8913-8928.

Lance, S., T. Raatikainen, et al. (2013). "Aerosol mixing state, hygroscopic growth and cloud activation efficiency during MIRAGE 2006." Atmospheric Chemistry and Physics **13**(9): 5049-5062.

Massoli, P., A. T. Lambe, et al. (2010). "Relationship between aerosol oxidation level and hygroscopic properties of laboratory generated secondary organic aerosol (SOA) particles." Geophysical Research Letters **37**.

Mei, F., A. Setyan, et al. (2013). "CCN activity of organic aerosols observed downwind of urban emissions during CARES." Atmospheric Chemistry and Physics **13**(24): 12155-12169.

Mochida, M., Nishita-Hara, C., Kitamori, Y., Aggarwal, S. G., Kawamura, K., Miura, K., and Takami, A.: Size-segregated measurements of cloud condensation nucleus activity and hygroscopic growth for aerosols at Cape Hedo, Japan, in spring 2008, J. Geophys. Res.: Atmos., 115, 2010.

Koehler, K., Kreidenweis, S., DeMott, P., Prenni, A., Carrico, C., Ervens, B., and Feingold, G.: Water activity and activation diameters from hygroscopicity data-Part II: Application to organic species, Atmos. Chem. Phys., 6, 795-809, 2006.

Pajunoja, A., Lambe, A. T., Hakala, J., Rastak, N., Cummings, M. J., Brogan, J. F., Hao, L., Paramonov, M., Hong, J., and Prisle, N. L.: Adsorptive uptake of water by semisolid secondary organic aerosols, Geophys. Res. Lett. 42, 3063-3068, 2015.

Petters, M., and Kreidenweis, S.: A single parameter representation of hygroscopic growth and cloud condensation nucleus activity, Atmos. Chem. Phys., 7, 1961-1971, 2007.

Petters, M. D., Wex, H., Carrico, C. M., Hallbauer, E., Massling, A., McMeeking, G. R., Poulain, L., Wu, Z., Kreidenweis, S. M., and Stratmann, F.: Towards closing the gap between hygroscopic growth and activation for secondary organic aerosol

– Part 2: Theoretical approaches, Atmos. Chem. Phys., 9, 3999-4009, 10.5194/acp-9-3999-2009, 2009.

Topping, D. O., McFiggans, G. B., and Coe, H.: A curved multi-component aerosol hygroscopicity model framework: Part 1 – Inorganic compounds, Atmos. Chem. Phys., 5, 1205-1222, 10.5194/acp-5-1205-2005, 2005.

Zhang, X. Y., Wang, J. Z., Wang, Y. Q., Liu, H. L., Sun, J. Y., and Zhang, Y. M.: Changes in chemical components of aerosol particles in different haze regions in China from 2006 to 2013 and contribution of meteorological factors, Atmos. Chem. Phys., 15, 12935-12952, 2015.

Zhao, D. F., A. Buchholz, et al. (2015). "Size-dependent hygroscopicity parameter ($\kappa$) and chemicalcomposition of secondary organic cloud condensation nuclei." Geophysical Research Letters **42**(24): 10920-10928.

---

## Author Response (AR2)

We would like to thank the referee #1 for providing further valuable comments on our manuscript and we have carefully addressed the referee's comments as follows (referee's comments in black and our responses in red):

1. For the CCNC calibration, I would suggest the authors provide the parameterizations of water activity and surface tension of ammonium sulfate, considering these are two major parameters in the Köhler theory.

Reply: In the CCNc calibration, the water activity ($a_w$) was approximated according to Rose et al. (2008),

$$a_w = \exp(-i_s \mu_s M_w)$$

where $i_s$, $\mu_s$, and $M_w$ is the van't Hoff factor, molality of solute, and the molar mass of water (0.01802 kg mol$^{-1}$), respectively. The van't Hoff factor $i_s$ is calculated from a polynomial fit to Pitzer model output data (Morre et al., 2010). The surface tension of the ammonium sulfate solution was approximated by the surface tension of pure water (0.072 N m$^{-1}$) according to Seinfeld and Pandis (2007), based on the fact that surface tension can only produce minor impacts on supersaturation.

We now incorporated several sentences to L18-22 on p. 10 and L1-3 on p. 11, "In the CCNc calibration, the water activity ($a_w$) was approximated according to Rose et al. (2008),

$$a_w = \exp(-i_s \mu_s M_w) \tag{3}$$

where $i_s$, $\mu_s$ and $M_w$ is the van't Hoff factor, molality of solute, and the molar mass of water (0.01802 kg mol$^{-1}$), respectively. The van't Hoff factor $i_s$ is calculated from a polynomial fit to Pitzer model output data (Morre et al., 2010). In this study, we adopted the simplest parameterization of the surface tension of the solution (Rose et al., 2008), that is, it was simply approximated by the surface tension of pure water (0.072 N m$^{-1}$) according to Seinfeld and Pandis (2007)".

2. Fig.9: the data seems dispersive but the R2 is 0.94, please recheck the data.

Reply: Per the reviewer's suggestion, we rechecked the data and the $R^2$ of the linear regression fit using different programs (e.g., Excel and Igor). It turned out that the $R^2$ value is still 0.94. By further digging into the literature, we found that data such as "Anscombe's quartet" (Anscombe, 1973) can have a high $R^2$ value induced by a few outliers while the other data points do not show a strong

correlation. As shown in Fig. 9, our measured data points were constrained within a small area (i.e. certain supersaturations or particle diameters). It turns out that the $R^2$ value is not a good indicator to represent the correlation between the measured and predicted AR in our case. Instead of using the $R^2$ value, we now introduced the p-value with a value less than 0.05 meaning significant correlation. Figure 9 shows a p-value close to 0, indicating a significant correlation between the measured and predicted AR. We have now modified Fig. 9 and added one sentence in L5-7 on P. 25, "We calculated the p-value between the measured and predicted AR and the results showed that the p-value is close to 0, indicating a significant correlation between the two variables."

[Figure]

3. Did the authors perform any water-soluble organic carbon (WSOC) measurements during the campaign? If so, maybe it is better to compare the kappa-org with fraction of WSOC.

Reply: The reviewer raised a very good point. Unfortunately, the WSOC species were not measured during the campaign and hence we cannot compare the kappa-org with the fraction of WSOC.

References:

Anscombe, F.J. (1973). Graphs is Statistical Analysis, Am. Stat., 27, 17-21.

Moore, R., Nenes, A., and Medina, J.: Scanning Mobility CCN Analysis—A Method for Fast Measurements of Size-Resolved CCN Distributions and Activation Kinetics, Aerosol Sci. Technol., 44, 861-871, 2010.

Rose, D., Gunthe, S., Mikhailov, E., Frank, G., Dusek, U., Andreae, M. O., and Pöschl, U.: Calibration and measurement uncertainties of a continuous-flow cloud condensation nuclei counter (DMT-CCNC): CCN activation of ammonium sulfate and sodium chloride aerosol particles in theory and experiment, Atmos. Chem. Phys., 8, 1153-1179, 2008.

We would like to thank the referee #2 for providing further valuable comments on our manuscript and we have carefully addressed the referee's comments as follows (referee's comments in black and our responses in red):

1. In the response to Comment 8, it is stated that "…the one between kAMS and kCCN became statistically significant…". The p value should be provided here.

   Reply: We calculated the p-value between the $\kappa_{AMS}$ and the $\kappa_{ccn}$. It turns out that it was close to 0, indicating that the correlation between the $\kappa_{AMS}$ and the $\kappa_{ccn}$ is significant. We have added one sentence in L20-21 on p.18, "The p-value between the $\kappa_{AMS}$ and the $\kappa_{ccn}$ was close to 0, indicating that the correlation between them is significant."

2. Regarding my Comment 12, I meant that the parameter C should be defined.

   Reply: The parameter C is only a fitting coefficient which does not have any specific physical meaning. The sentence in L3-5 on p.14 has been revised, "Note that the parameter C is a fitting coefficient with no specific physical meaning. However, a small C value indicates a steep activation curve."

3. In the new Figure 1, the legend in the inset of activation ratio vs Dp is blurred.

Reply: We used "AR" instead of "Activation Ratio" in legend. The figure has been revised, as shown below:

[Figure]